# Giant nonlinear Hall and wireless rectification effects at room temperature in the elemental semiconductor tellurium

Bin Cheng[1,2,3,6], Yang Gao ®[1,2,6], Zhi Zheng[1,2,3,6], Shuhang Chen[4], Zheng Liu[2], Ling Zhang[1,2,3], Qi Zhu[4], Hui Li ®[5], Lin Li ®[1,2,3] ✉ & Changgan Zeng ®[1,2,3] ✉

The second-order nonlinear Hall effect (NLHE) in non-centrosymmetric materials has recently drawn intense interest, since its inherent rectification could enable various device applications such as energy harvesting and wireless charging. However, previously reported NLHE systems normally suffer from relatively small Hall voltage outputs and/or low working temperatures. In this study, we report the observation of a pronounced NLHE in tellurium (Te) thin flakes at room temperature. Benefiting from the semiconductor nature of Te, the obtained nonlinear response can be readily enhanced through electrostatic gating, leading to a second-harmonic output at 300 K up to 2.8 mV. By utilizing such a giant NLHE, we further demonstrate the potential of Te as a wireless Hall rectifier within the radiofrequency range, which is manifested by the remarkable and tunable rectification effect also at room temperature. Extrinsic scattering is then revealed to be the dominant mechanism for the NLHE in Te, with symmetry breaking on the surface playing a key role. As a simple elemental semiconductor, Te provides an appealing platform to advance our understanding of nonlinear transport in solids and to develop NLHE-based electronic devices.

As a new member of the Hall family, the nonlinear Hall effect (NLHE) has recently sparked widespread attention as it can exist in time-reversal invariant systems. It greatly broadens the horizon of Hall effect studies, transcending the constraints of Onsager reciprocity relation[1–4]. In contrast to conventional Hall effects in the linear response regime, NLHE is generated as a second-order electrical response to an applied alternating current (AC) with a certain frequency ($\omega$) and can give rise to Hall voltages with second-harmonic ($2\omega$) and zero frequencies without introducing an external magnetic field[2,5]. Initially, NLHE was proposed to arise from the Berry curvature dipole (BCD) in non-magnetic materials[1]. That is, inversion-symmetry breaking may segregate the positive and negative Berry curvatures in different momentum regions, giving rise to a dipole moment when the crystalline symmetry allows. Therefore, topological materials with tilted Dirac or Weyl cones, which serve as sources of large BCD, have emerged as ideal candidates for realizing NLHE[1,6]. Indeed, experimental evidence of NLHE has been observed in several topological semimetals, such as bilayer WTe$_2$[2] and TaIrTe$_4$[7], providing validation for this intrinsic mechanism. On the other hand, some materials have recently been reported to exhibit NLHE without BCD, which is represented by certain two-dimensional (2D) artificial systems like graphene/BN moiré superlattices[8] and twisted bilayer graphene[9].

[1]CAS Key Laboratory of Strongly-Coupled Quantum Matter Physics, and Department of Physics, University of Science and Technology of China, Hefei, Anhui 230026, China. [2]International Center for Quantum Design of Functional Materials (ICQD), Hefei National Research Center for Physical Sciences at the Microscale, University of Science and Technology of China, Hefei, Anhui 230026, China. [3]Hefei National Laboratory, University of Science and Technology of China, Hefei, Anhui 230088, China. [4]Department of Electronic Engineering and Information Science, University of Science and Technology of China, Hefei, Anhui 230026, China. [5]Institutes of Physical Science and Information Technology, Anhui University, Hefei, Anhui 230601, China. [6]These authors contributed equally: Bin Cheng, Yang Gao, Zhi Zheng. ✉e-mail: lilin@ustc.edu.cn; cgzeng@ustc.edu.cn

This BCD-free NLHE has been attributed to some extrinsic mechanisms like disorder-related scattering[5,10], which further enriches our understanding of such a nonlinear transport phenomenon.

Besides its scientific significance in investigating quantum geometry and crystalline symmetry, NLHE has extensive application prospects in the area of frequency-doubling and rectifying devices. In particular, such NLHE can also be achieved by replacing the driving AC current with an oscillating electromagnetic field, enabling its utilization as a wireless rectifier[11,12]. Unlike conventional rectifiers that rely on the fabrication of p-n junctions or metal-semiconductor junctions, such a type of Hall rectifier is based on the inherent property of the material and thus avoids limitations posed by transition time and thermal voltage threshold[13,14]. Therefore, it is highly promising to achieve a NLHE-based wireless rectifier with broadband response under zero bias, which holds great potential for applications in energy harvesting and wireless charging.

Despite the rapid progress in both aspects of material discovery and mechanism elucidation, previously reported NLHE is always plagued by small Hall outputs and/or low operating temperatures, thus hindering the development of NLHE-based new-principle electronic devices[2,7–9,15–23]. To our best knowledge, room temperature (RT) NLHE has been observed exclusively in the Dirac semimetal BaMnSb$_2$[23] and on the surface of the Weyl semimetal TaIrTe$_4$[7], wherein the intrinsic BCD mechanism holds. However, the obtained voltage outputs for both the NLHE and the NLHE-based wireless radiofrequency (RF) rectification are relatively small. Moreover, the inherent lack of tunability in such semimetal systems limits their further enhancement. It is thus of vital importance to seek out a material system with superior NLHE performance. In this work, we systematically investigate the nonlinear transport in thin flakes of elemental semiconductor tellurium (Te). A giant and tunable NLHE is achieved at RT, and the device application for wireless rectification is further demonstrated.

## Results

### RT NLHE in Te thin flakes

Te is an elemental semiconductor with a narrow bandgap of ~0.38 eV[24,25]. It is composed of one-dimensional atom helical chains along the c-axis (Fig. 1a), whose inversion symmetry is broken (see Supplementary Fig. 1). Quasi-2D Te flakes used for the NLHE measurements were grown via a simple hydrothermal method following previous reports[26,27]. These as-grown Te flakes typically exhibit a trapezoidal shape, while the directions parallel and perpendicular to their long edges are the c-axis and a-axis, respectively. We first checked the in-plane resistance anisotropy (using two terminals) in a circular disc device (labeled with #D1), where the angle between the AC current $I^\omega$ and the c-axis is denoted as $\theta$ (Fig. 1b). As shown in Fig. 1c, the first-harmonic longitudinal resistance $R_{xx}$ demonstrates a twofold angular dependence. $R_{xx}$ reaches its minimum (maximum) when $\theta = 0°$ and 180° (90° and 270°). This observation can be reasonably attributed to the one-dimensional atomic chain structure of Te, as carriers move more easily along the atomic chains (c-axis).

NLHE measurements were then conducted on the same disc device using the standard lock-in technique. In these measurements, a pair of electrodes was selected to apply $I^\omega$, and the Hall voltage was collected in the perpendicular direction without applying a magnetic field. Figure 1d presents the results for several typical current directions measured at 300 K, from which sizeable outputs of second-harmonic Hall voltage ($V_{xy}^{2\omega}$) are clearly seen. Though the amplitude varies with $\theta$, all the measured $V_{xy}^{2\omega}$ exhibit a well-defined quadratic dependence on the applied current $I^\omega$. To further explore the in-plane anisotropy, $V_{xy}^{2\omega}$ versus $I^\omega$ curves for additional angles were measured in the same way, and the obtained nonlinear Hall voltages for $I^\omega = 51\,\mu A$ are plotted as a function of $\theta$ in Fig. 1e. Notably, $V_{xy}^{2\omega}$ exhibits a onefold angular dependence, with its maximum achieved at $\theta = 90°$ and 270°, where the current is parallel to the a-axis. Similar angular dependence

is also observed in other Te devices, indicating a good reproducibility (see Supplementary Fig. 5).

We then conducted systematic measurements on the nonlinear response for $\theta = 270°$. From Fig. 1f, a second-harmonic signal in the longitudinal direction ($V_{xx}^{2\omega}$) is also observed, with its amplitude comparable to that of $V_{xy}^{2\omega}$. This result is distinctly different from the linear transport data, wherein the first-harmonic Hall voltage is negligible in comparison to the longitudinal one (see Supplementary Fig. 2a). We note that for the intrinsic NLHE contributed by the BCD (**D**), the as-obtained nonlinear Hall current $\mathbf{J}_{NLHE}$ can be expressed as $\mathbf{J}_{NLHE} \propto (\mathbf{D} \cdot \mathbf{E})\hat{\mathbf{z}} \times \mathbf{E}$, where **E** is the applied in-plane longitudinal electric field and $\hat{\mathbf{z}}$ represents the out-of-plane direction[15]. As a result, the corresponding second-order response occurs in the transverse direction that is perpendicular to **E**, and no signal develops in the longitudinal direction. Here, the simultaneous observation of $V_{xy}^{2\omega}$ and $V_{xx}^{2\omega}$ with comparable amplitudes indicates that a scattering-related extrinsic mechanism, rather than the intrinsic BCD, should play a leading role in generating NLHE in Te[10,11]. Similar phenomena have already been reported in other NLHE systems, represented by graphene/BN superlattice[8] and twisted bilayer graphene[9].

Following the expression $V_{xy} \propto (I_0 \sin(\omega t))^2 = I_0^2(1 + \sin(2\omega t - \pi/2))/2$, the second-order NLHE can generate a direct-current (DC) voltage output in the Hall direction in addition to the second-harmonic one, which offers an alternative route to realize current rectification[2,5]. This is indeed what we achieved in the Te devices. As demonstrated in Fig. 1g, a DC Hall voltage $V_{xy}^{DC}$ that scales quadratically with the applied AC current is seen, which exhibits an amplitude almost identical to the second-harmonic component as expected. To further validate our experimental findings, we conducted Hall measurements using input currents with different frequencies, and the as-obtained second-harmonic signal is independent of the frequency (see Fig. 1h). In addition, the signs of both $V_{xy}^{DC}$ and $V_{xy}^{2\omega}$ remain unchanged when the direction of the driving current switches (see Supplementary Fig. 2b). All these observations collectively confirm the presence of a second-order nonlinear electrical response in Te flakes at RT. Other possible origins, like diode effect, thermoelectric effect, etc., have also been carefully excluded (as detailed in Supplementary Note 4).

### Highly-tunable and giant NLHE in Te devices

In addition to a high working temperature, achieving a relatively large output is another key issue for practical applications based on NLHE. Previous studies have demonstrated that the strength and even the polarity of NLHE present a strong correlation with the nature of the Fermi surface, which governs the scattering characteristics and the distribution of the BCD[1,10]. For narrow-bandgap semiconductors like Te, the relatively low carrier density enables effective tuning of the Fermi level via electrostatic gating, which has been well demonstrated in the transport performance within the linear regime[28,29]. We thus proceed to further tune the nonlinear transport in our Te devices by applying a back-gate voltage ($V_{BG}$) while utilizing SiO$_2$ as the dielectric layer.

We first measured the variation of the first-harmonic resistivity $\rho_{xx}$ while sweeping $V_{BG}$. From the results shown in Supplementary Fig. 3b, $\rho_{xx}$ increases with increasing $V_{BG}$, indicating a pristine hole-doping characteristic of Te flake[26,29]. Subsequently, the nonlinear transport data under different $V_{BG}$ are obtained, as plotted in Fig. 2a. For all the measured $V_{BG}$, a remarkable signal of $V_{xy}^{2\omega}$ that scales linearly with $(I^\omega)^2$ was obtained, and the amplitude increases monotonically as $V_{BG}$ increases. To better demonstrate the tuning effect, we plotted the extracted $V_{xy}^{2\omega}$ for the maximum input current ($I^\omega = 51\,\mu A$) as a function of $V_{BG}$ in Fig. 2b. Over the range of applied $V_{BG}$, $V_{xy}^{2\omega}$ is tuned by more than two orders of magnitude, ranging from $1.2 \times 10^1\,\mu V$ at $V_{BG} = -60\,V$ to $2.8 \times 10^3\,\mu V$ at $V_{BG} = 45\,V$ (see the logarithmic plot in the inset).

In Fig. 2d, we present a graphical representation of the $V_{xy}^{2\omega}$ maximum and corresponding operating temperatures for typical previously

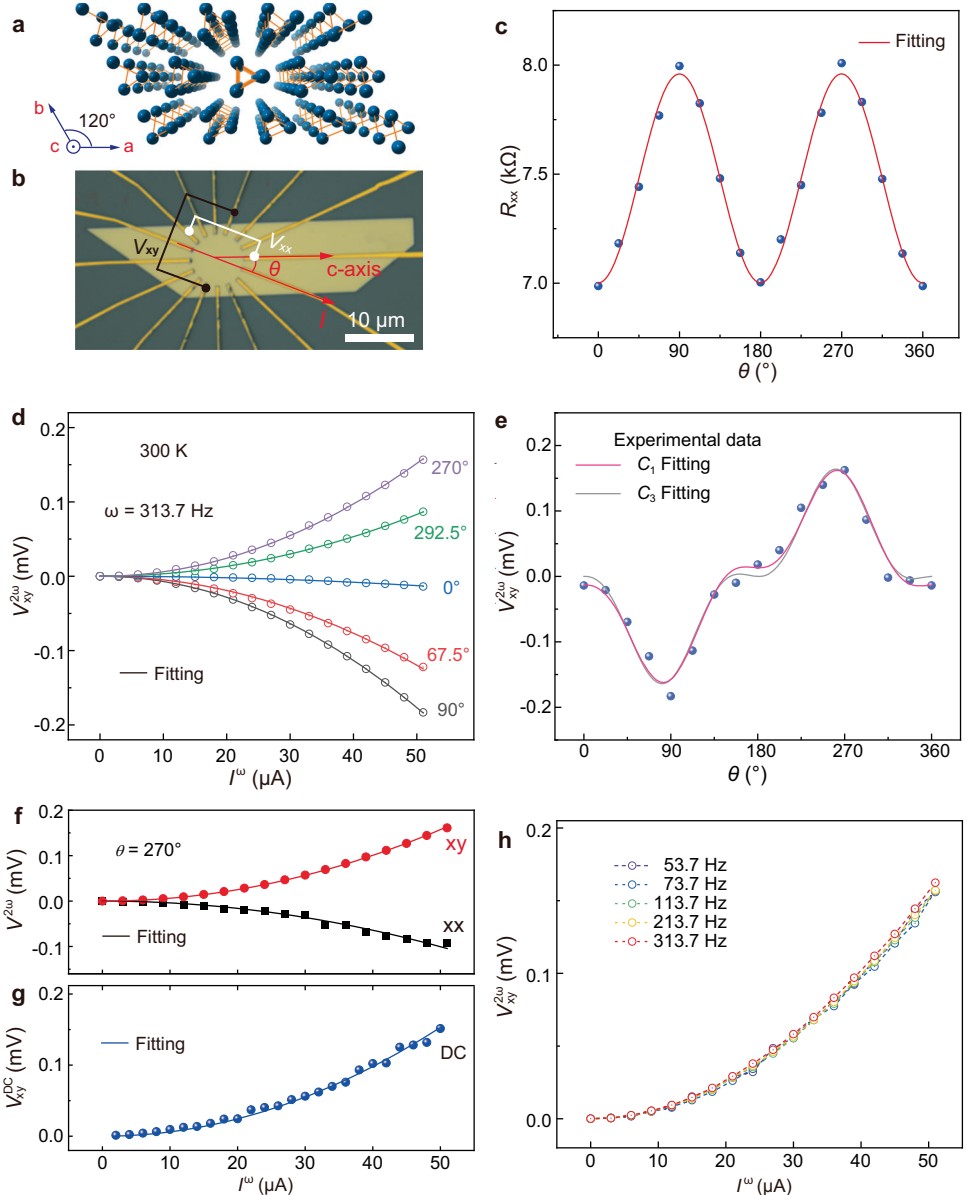

**Fig. 1 | Room-temperature nonlinear Hall effect (NLHE) in Te device. a** Crystal structure of Te. **b** Optical image of the circular disc device #D1. Here, $\theta$ is the angle between the input AC current ($I^\omega$) and the $c$-axis of Te flake. $V_{xx}^{2\omega}$ and $V_{xy}^{2\omega}$ represent the measured second-harmonic longitudinal and Hall voltages, respectively. **c** Longitudinal resistance $R_{xx}$ as a function of $\theta$, which can be well fitted by $R_{xx}(\theta) = R_c\cos^2(\theta) + R_a\sin^2(\theta)$. Here, $R_a$ and $R_c$ are the resistance along the $a$-axis and $c$-axis, respectively. **d** $V_{xy}^{2\omega}$ as a function of $I^\omega$ at several typical angles. The solid lines are the quadratic fitting results. The frequency of $I^\omega$ is 313.7 Hz. **e** Angular dependence of $V_{xy}^{2\omega}$ for $I^\omega = 51\,\mu A$. Solid lines are corresponding fitting results by using the as-deduced equations for two different cases of surface symmetry breaking. $C_3$ Fitting (grey line): only $C_2$ symmetry is broken, and the point group is reduced to $C_3$; $C_1$ Fitting (pink line): both $C_3$ and $C_2$ symmetries are broken, and the point group is reduced to $C_1$. **f** $V_{xx}^{2\omega}$ and $V_{xy}^{2\omega}$ as functions of $I^\omega$ for $\theta = 270°$. **g** Corresponding DC Hall voltage as a function of $I^\omega$. The solid lines are the quadratic fitting results. **h** $V_{xy}^{2\omega}$ vs $I^\omega$ curves measured under various frequencies of $I^\omega$ for $\theta = 270°$. All data in Figs. 1–4 were taken at 300 K unless otherwise noted.

reported NLHE systems. It is evident that most systems exhibit significant NLHE at relatively low temperatures, normally below 10 K[2,8,9,15–22]. The achievement of RT NLHE has been limited to specific semimetal systems, namely, Dirac semimetal BaMnSb$_2$[23] and Weyl semimetal TaIrTe$_4$[7]. We note that BaMnSb$_2$ holds the record for the highest RT nonlinear Hall output, i.e., $2.5 \times 10^2\,\mu V$ when injecting a higher AC current of 100 μA. The maximum $V_{xy}^{2\omega}$ achieved in our Te device ($2.8 \times 10^3\,\mu V$) surpasses this record by one order of magnitude, which is benefited from its semiconductor nature. In addition, the data of power efficiency[11] for typical systems were also estimated, which further demonstrate the superior NLHE performance of Te (see Supplementary Fig. 6).

In addition to the electrostatic gating, the temperature effect on the NLHE performance has also been examined, and the results obtained at temperatures from 300 K to 200 K are shown in Fig. 2c. It is evident that $V_{xy}^{2\omega}$ scales linearly with $(I^\omega)^2$ at all the measured temperatures and the output value increases monotonically as the temperature decreases. The inset of Fig. 2c further illustrates the $V_{BG}$ dependent $V_{xy}^{2\omega}$ at 200 K, which was obtained by sweeping the $V_{BG}$ with a low input current of 10 μA. The achieved tuning effect is much more remarkable than that at 300 K, and the maximum value of $V_{xy}^{2\omega}$ reaches an impressive value of 148 mV, which is nearly two orders of magnitude larger than the 300 K one. As demonstrated in Fig. 2d, this output value

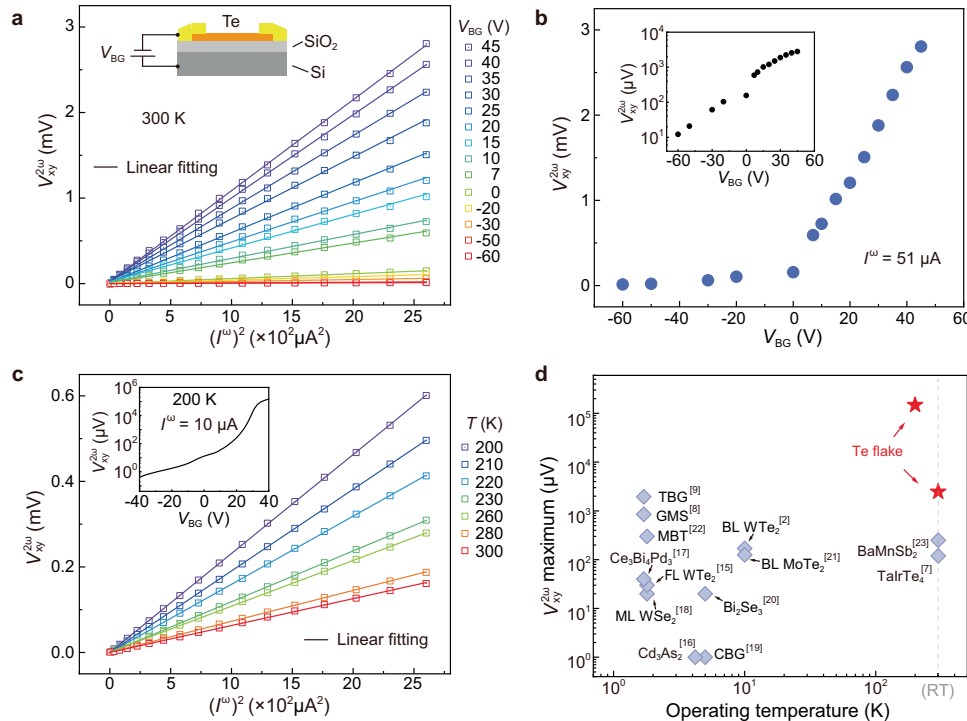

**Fig. 2 | Gate-tunable NLHE. a** $V_{xy}^{2\omega}$ as a function of $(I^\omega)^2$ measured at different back-gate voltages ($V_{BG}$). Inset: schematic of gate-voltage tuning in device #D1, wherein $V_{BG}$ is applied through $SiO_2$/Si substrate. **b** The obtained values of $V_{xy}^{2\omega}$ for a fixed $I^\omega = 51\,\mu A$ under different $V_{BG}$. Inset: corresponding logarithmic plot. **c** $V_{xy}^{2\omega}$ as a function of $(I^\omega)^2$ measured at different temperatures. Inset: $V_{xy}^{2\omega}$ vs $V_{BG}$ curve measured at 200 K with $I^\omega = 10\,\mu A$. **d** Comparison of the NLHE performance between our Te flakes and other typical systems reported previously, including natural materials like $WTe_2$ and $TaIrTe_4$, artificial structures represented by twisted bilayer graphene, etc. BL $WTe_2$: bilayer $WTe_2$, GMS: graphene moiré superlattice, MBT: $MnBi_2Te_4$, BL $MoTe_2$: bilayer $MoTe_2$, FL $WTe_2$: few-layer $WTe_2$, ML $WSe_2$: monolayer $WSe_2$, TBG: twisted bilayer graphene, CBG: corrugated bilayer graphene. Solid lines in (**a**) and (**c**) are linear fitting results. All the measurements were conducted with the applied current along the $a$-axis.

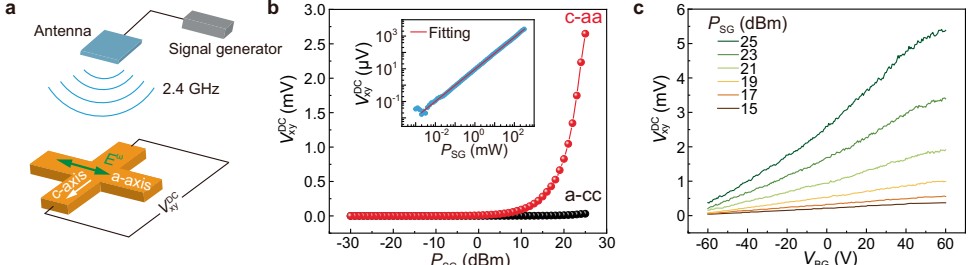

**Fig. 3 | Wireless radiofrequency (RF) rectification based on NLHE. a** Schematic of the measurement setup. RF signal is generated by a signal generator with a frequency of 2.4 GHz, and a patch antenna is employed as a radiation source. Here, $\mathbf{E}^\omega$ is the oscillating electric field of RF signal, and $V_{xy}^{DC}$ represents the rectified DC Hall voltage. **b** $V_{xy}^{DC}$ as a function of signal power $P_{SG}$ measured in device #C1. Here, the c-aa (a-cc) means that the direction of the oscillating electric field is set along the $a$-axis ($c$-axis), while $V_{xy}^{DC}$ is collected along the perpendicular direction, i.e., $c$-axis ($a$-axis). Inset: corresponding logarithmic plot and power-law fitting result for the c-aa case. **c** $V_{BG}$ dependent $V_{xy}^{DC}$ for various $P_{SG}$.

significantly surpasses those achieved in previous systems, including the ones presenting NLHE at much lower temperatures.

## Wireless rectification based on NLHE

As mentioned earlier, the second-order NLHE can be utilized to convert oscillating electromagnetic fields into a DC voltage without external bias, thus enabling a type of "Hall rectifier"[7,12]. Motivated by the observation of giant RT NLHE in our devices, we further explore the potential of Te thin flakes in realizing wireless RF rectification. Figure 3a shows the schematic of the experimental setup, among which a cross-like Te device (labeled with #C1) was fabricated with its edges aligned along the $a$-axis and $c$-axis, respectively. RF signal is radiated onto the device using a patch antenna with a central frequency of ~2.4 GHz, which corresponds to

the frequency of a common Wi-Fi channel. Figure 3b illustrates the rectified output $V_{xy}^{DC}$ with varying the power of signal generator $P_{SG}$. When the direction of the oscillating electric field $\mathbf{E}^\omega$ is oriented along the $a$-axis, obvious rectified voltage can be obtained along the $c$-axis (c-aa) as $P_{SG}$ increases above a certain value of −27 dBm. In contrast, $V_{xy}^{DC}$ remains negligible when $\mathbf{E}^\omega$ is along the $c$-axis (a-cc), which agrees with the result of angular dependent NLHE shown in Fig. 1e. As demonstrated in the inset of Fig. 3b, the corresponding logarithmic plot of $V_{xy}^{DC}$ versus $P_{SG}$ data for the c-aa case follows a simple power law $V_{xy}^{DC} \propto (P_{SG})^\alpha$ with $\alpha = 0.98$. Such a well-defined linear power dependence of rectified voltage indicates a second-order nonlinear response, consistent with its NLHE origin[30]. Similar rectification performance is also observed in another cross-like device #C2 (see Supplementary Fig. 7).

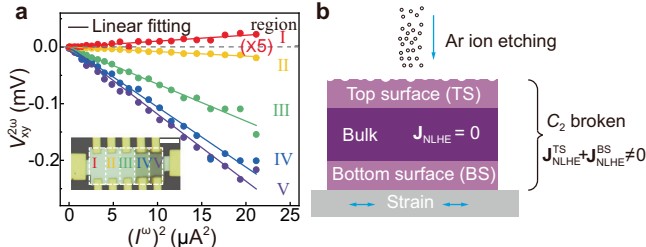

**Fig. 4 | Etching effect on the performance of NLHE in Te. a** $V_{xy}^{2\omega}$ versus $(I^\omega)^2$ curves for different regions in the Hall bar device #H1. Inset: Optical image of the device with its channel along the $a$-axis. The scale bar is 5 μm. The thickness of regions I–V (from left to right) is 32 nm, 28 nm, 22 nm, 15 nm, 12 nm, respectively. The measured $V_{xy}^{2\omega}$ collected on region I is amplified by five times for better comparison. **b** Schematic of NLHE contributions from different parts of Te flake. Here $\mathbf{J}_{NLHE}$ is the total nonlinear Hall current density, $\mathbf{J}_{NLHE}^{TS}$ and $\mathbf{J}_{NLHE}^{BS}$ are the contributions from the top- and bottom surface, respectively. The breaking of the $C_2$ symmetry on the surface, which can be induced by either substrate strain or surface etching, leads to a nonzero nonlinear Hall signal.

This RT wireless rectification effect can also be readily modified via applying $V_{BG}$. As demonstrated in Fig. 3c, $V_{xy}^{DC}$ increases monotonically as $V_{BG}$ increases from −60 V to 60 V for all applied $P_{SG}$. The maximum $V_{xy}^{DC}$ reaches 5.4 mV at $V_{BG}$ = 60 V with $P_{SG}$ set to be 25 dBm. This value is much higher than those achieved in other systems presenting NLHE-based RF rectification[7,22], e.g., two orders of magnitude larger than the one achieved in TaIrTe$_4$[7] (60 μV, 300 K).

Additionally, we have checked the rectification performance of our Te devices by varying the frequency of the RF signal, and significant response can be achieved when the frequency ranges from 300 MHz to 4.5 GHz (see Supplementary Fig. 8). All these observations establish Te as an efficient and highly-tunable Hall rectifier that can work at RT and over a broad RF range. If further optimizing the device, for example, by electrically connecting it to a receiving antenna to improve the collection efficiency of RF signal radiation[31], it will be highly promising to achieve superior rectification performance.

## Possible mechanism of NLHE in Te

Below we would like to explore the possible origin for the NLHE in our Te devices. As mentioned above, the coexistence of nonlinear signals in both the longitudinal and transverse directions with comparable amplitudes points towards a scattering-dominated extrinsic mechanism[10,11]. This is further supported by the scaling law analysis (see Supplementary Note 3). However, for either intrinsic or extrinsic mechanisms, the crystalline-symmetry constraint for realizing NLHE is quite stringent[5]. Te crystal belongs to the $D_3$ point group, whose symmetric operations include a threefold rotation $C_3$ about the $c$-axis and twofold rotation $C_2$ about the $a$-axis. Such a crystalline symmetry, however, does not allow the emergence of a nonlinear Hall signal (see Supplementary Note 1). One possible explanation hinges on the reduction of symmetry near the surface, a common occurrence in low-dimensional systems[9,32–34], leading to a nonzero nonlinear Hall current near the top and bottom surfaces. As detailed in Supplementary Note 1, we have examined three possible scenarios for the surface symmetry breaking: (I) The $C_2$ symmetry is broken, (II) The $C_3$ symmetry is broken, and (III) The $C_2$ and $C_3$ symmetries are both broken. For either case I or III, the reduced symmetry allows the emergence of a nonlinear Hall signal along the principal axes. The angular-dependent nonlinear Hall data can be adequately described by the equations derived from the corresponding symmetry analyses, while the equation deduced from the case III provides a better description (see Fig. 1e). However, if we solely consider the breaking of $C_3$ symmetry on the surface (case II), the residual $C_2$ symmetry prohibits the appearance of nonlinear Hall response along the $c$-axis with the applied current along the $a$-axis.

Consequently, the breaking of $C_2$ symmetry on the surface is the minimal requirement to account for the observed nonlinear Hall response in Te flakes. Such surface symmetry breaking may be due to the fact that the atoms in the 1D helical chain experience different atomic environment on the surfaces, and the substrate may further induce additional tension onto the bottom surface.

Such surface effect is further verified in our subsequent experiments on the as-grown Te flakes with different thicknesses (as detailed in Supplementary Note 5), and also the etched samples. For the latter case, a Hall bar device (labeled with #H1) is used and the channel comprises regions with different thickness (see inset of Fig. 4a). Except for region I, the other four regions were thinned via Ar ion etching, giving rise to a gradual reduction of thickness from section I to V. As Fig. 4a shows, NLHE is observed when detecting the voltage drop between Hall electrodes in all regions. Importantly, a sign reversal of the second-harmonic output $V_{xy}^{2\omega}$ is clearly observed between the pristine region I and the other four etched regions. Such etching-induced polarity reversal is also observed in another Hall bar device #H2, demonstrating its reproducibility (see Supplementary Fig. 11). As the etching only changes the surface roughness and the sample thickness while leaving the bulk intact, such a sign reversal clearly demonstrates that the surface contribution plays a vital role (see the schematic in Fig. 4b).

The origin of the sign reversal resides deeply in the scaling behavior of the nonlinear Hall signal. As detailed in Supplementary Note 2, the general scaling equation between the nonlinear Hall strength and the sample resistivity can be reduced to the following form: $E_{xy}^{2\omega}/(E_{xx}^\omega)^2 = a' + b'/\rho_{xx}$, where $E_{xy}^{2\omega} = V_{xy}^{2\omega}/W$ and $E_{xx}^\omega = V_{xx}^\omega/L$, $L$ and $W$ are the length and width of the channel, respectively. We note that $a' = C^{in} + C_0^{sj} + C_{00}^{sk,1}$ is a mixing of the intrinsic, side-jump, and Gaussian-type skew-scattering contribution, and $b' = C^{sk,2}$ is due to non-Gaussian type skew scattering[9,10,20]. The sign reversal then naturally appears when a' and b' have different signs. We shall note that such coefficients with opposite signs have also been observed in the study of the anomalous Hall effect in the iron films[35]. But the intrinsic contribution in that case dominates and the sign reversal requires a large conductivity. In comparison, in our case the sign reversal can occur with a moderate conductivity, suggesting a large non-Gaussian type skew-scattering contribution.

In summary, we observed a giant NLHE at RT in the elemental semiconductor Te and further demonstrated its application as a wireless "Hall rectifier" under different RF frequencies. Furthermore, both the performance of NLHE and NLHE-based RF rectification can be notably improved by electrostatic gating, giving rise to a voltage output significantly surpassing those of previously reported systems. Further investigations have revealed that the giant RT NLHE is dominated by a scattering–related extrinsic mechanism and is rooted in the reduction of crystalline symmetry on the surface. These findings establish Te as an appealing system for exploring NLHE and related phenomena, which will definitely deepen our comprehension of nonlinear transports and, moreover, advance the field of nonlinear electronic devices. On the other hand, Te has already exhibited a wealth of fascinating behaviors in the linear transport regime, like chiral anomaly[28,36], quantum Hall effect[37,38], and nonreciprocal charge-to-spin conversion[39]. Here, the discovery of its good performance in the nonlinear transport regime will undoubtedly stimulate substantial enthusiasm for further exploring the exotic electronic properties of this elemental semiconductor.

## Methods
### Device fabrication
Te flakes were synthesized using the hydrothermal method[26,27]. 46 mg of Na$_2$TeO$_3$ and 1.5 g of polyvinylpyrrolidone were first dissolved in 16 ml of deionized water with continuous stirring. After that, ammonium hydroxide and hydrazine monohydrate were added into the

resulted solution. The final solution was then sealed in a Teflon-lined stainless-steel autoclave and maintained at 180 °C for 10 h. After cooling down naturally, the obtained product was washed using deionized water to remove residual ions.

The as-grown Te flakes usually have a trapezoid shape, with typical thickness ranging from 20 to 100 nm. For gate-tuning experiments, Te flakes with relatively small thickness of 20–30 nm were selected. Before device fabrication, the high quality of the as-grown Te samples was carefully checked by using high-resolution transmission electron microscope and Raman spectrum, and the typical results are shown in Supplementary Fig. 1. To fabricate the disc devices, Te flakes were transferred onto a 285 nm $SiO_2$/Si substrate. Sixteen electrodes were defined into a circular shape using electron beam lithography, and 0.5/20/70 nm Ti/Pd/Au were then deposited as contact metals using e-beam evaporation. For the Hall-bar and cross-like devices, similar fabrication procedures were adopted after etching the Te flakes into corresponding shapes using Ar ion beam etching.

## Transport measurements

For the NLHE measurements, AC current was applied to the devices by using a Keithley 6221 current source. First- and second-harmonic longitudinal/transverse voltages were measured using lock-in amplifiers (SR830). Rectified DC voltage was collected using a Keithley 2182A voltmeter. In our experiments, five different driving frequencies (53.7 Hz, 73.7 Hz, 113.7 Hz, 213.7 Hz, 313.7 Hz) were tested, and the results are independent of the frequency, confirming the validity of our findings.

To carry out the RF rectification measurements, RF signals were generated using a signal generator (E8257D, Agilent), and an antenna was used as a wireless RF radiation source, which was placed 1 cm away from the Te devices during the measurements. Three types of antenna (patch antenna, dipole antenna, and log-periodic antenna) were used in our experiments. Corresponding results are respectively shown in Fig. 3 and Supplementary Figs. 7 and 8, from which similar RF rectification behaviors are clearly seen.

## Data availability

The data represented in Figs. 1, 2, 3, and 4 are available as Source Data files. All other data that support the plots within this paper and other findings of this study are available from the corresponding author on request. Source data are provided with this paper.

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

## Acknowledgements

This work was supported by the CAS Project for Young Scientists in Basic Research (Grant No. YSBR-046), the Anhui Provincial Natural Science Foundation (Grant No. 2308085J11), the National Natural Science Foundation of China (Grant Nos. 92165201 and 12234017), the National Key Research and Development Program of China (Grant No. 2023YFA1406300), the Anhui Provincial Key Research and Development Project (Grant No. 2023z04020008), the Innovation Program for Quantum Science and Technology (Grant No. 2021ZD0302800), the Fundamental Research Funds for the Central Universities (Grant Nos. WK2310000104 and WK2340000102). L.L. was also supported by USTC Tang Scholar. Part of this work was carried out at the Center for Micro- and Nanoscale Research and Fabrication, and the Instruments Center for Physical Science, University of Science and Technology of China.

## Author contributions

C.Z. and L.L. designed and supervised the work; B.C., Z.Z. performed the experiments with assistance from S.C., L.Z., Q.Z., and H.L.; Y.G. and Z.L. provided the theoretic support. L.L., B.C., Y.G., and C.Z. analyzed the data and wrote the manuscript; All authors contributed to the scientific discussion and manuscript revisions.

## Competing interests

The authors declare no competing interests.
