## [Peer Review File · Nature Communications]

Giant nonlinear Hall and wireless rectification effects at room temperature in the elemental semiconductor telluriumEditorial Note: Parts of this Peer Review File have been redacted as indicated to remove third-party material where no permission to publish could be obtained.

REVIEWER COMMENTS

Reviewer #1 (Remarks to the Author):

The manuscripts by Cheng et al. has demonstrated a strong room-temperature nonlinear Hall effect, and its rectification ability of microwave signal. I think this is an important work both in fundamental physics and novel electronic applications. It demonstrates the universality of geometric effects of electrons in solids. The wireless rectification effects could also enable potential useful applications.

The authors have done a careful job in analyzing the nonlinear Hall effect, including studying the current, angle and Fermi energy dependence. The data sufficiently supports their conclusion.

I only have one minor question about the angular dependence of the nonlinear Hall effects. I wonder if the authors have etched the devices to make the current flow along the targeted directions. It looks like the device doesn't have a rotational-symmetric shape. I understand it may not affect the observation, but have the authors done some finite-element simulation to support their observation if they haven't etched the device?

After all, I would like to recommend the papers for publication.

Reviewer #2 (Remarks to the Author):

In the manuscript, Cheng and co-authors present an impressive study of giant room temperature second-order Hall nonlinearity in 2D Te thin flakes. As for application, the tunability of nonlinear response and the wireless rectification effects in the GHz frequency band are clearly demonstrated. Although the physics origins related to quantum metric and Berry curvature dipole are fascinating, the simultaneous observation of longitudinal and Hall nonlinear voltages in this work points towards the extrinsic scattering mechanism. The control experiments and scaling fitting verify the skew-scattering origins. This work provides an attractive candidate in nonlinear Hall family and offers an essential step to the practical application. Therefore, I recommend this paper for publication in Nature Communications.

Several questions for further improvement are listed as follows:

In the manuscript, the crystal structure and the principal axis of Te are clearly shown in Fig. 1. The b-axis is not perpendicular to the a-axis and also the a-c plane. Indeed, it reflects directly on the symmetry analysis in Supplementary Note 1. Specifically, the y-axis is not perpendicular to the x-z plane as set by the authors in Line 28 to Line 30 in Supplementary Information. Thus, the symmetry analysis and discussion in Supplementary Note 1 should be more complicated in principle. I wonder if the authors have taken this crucial point in consideration.

It would be better if the characterization of the as-grown Te samples by high-resolution transmission electron microscope and Raman spectrum can be added into the manuscript or supplementary information. Specially, the polarized Raman spectra are needed to directly certify the angular-dependent transport results.

The two-fold rotation C2 about the a-axis is supposed to be broken at the surface but the three-fold rotation C3 about the c-axis is preserved. However, the three rows of Te atoms in the 1D triple helix chain at the surface experience different atomic environments. The C3 symmetry could be broken at the surface in principle.

The results of angular-dependent measurements of the second-order Hall voltage $V_{xy}^{2\omega}$ have been fitted in the Fig.1d using the equation in the Supplementary Information. It would be better if the authors add the angular-dependent fitting results of the observed longitudinal second-order voltage $V_{xx}^{2\omega}$ measured as shown in Fig.1e in the manuscript or supplementary information. Besides, if the analysis in Supplementary Note 1 is correct, the fitting coefficients of the Hall results and longitudinal results could be similar. Moreover, the fitting results of $V_{xx}^{2\omega}$ could help to eliminate the thermoelectric effect more persuasively. Because the Joule heating mainly contributes to the longitudinal transport.

As shown in Supplementary Fig.1a, the first-harmonic measurements show the Hall voltage is not zero. I think the reason should be clarified in the manuscript or supplementary information. If it originates from the unavoidable misalignment, it may also be another origin of the observed nonlinear Hall voltage or longitudinal voltage.

The two results from up and down channel of Supplementary Fig.1b are different. The 2ω and DC results are more identical in the down channel but the results in the up channel are not.

The thickness-dependent measurements are lacked for proof of the surface effect discussed in the Line 184 to Line 196 in manuscript.

The reported results (Q. Fu, et al. Berry Curvature Dipole Induced Giant Mid-Infrared Second-Harmonic Generation in 2D Weyl Semiconductor. *Adv. Mater.* 2306330 (2023)) show the Berry curvature dipole induced nonlinear optical physics in 2D Te. Due to the broken inversion symmetry of 2D Te, the Berry curvature dipole exists along particular directions. Could such non zero Berry curvature dipole affect the measured nonlinear transport phenomena?

Lines 40-41: “That is, inversion-symmetry breaking may segregate the positive and negative Berry curvatures in different momentum regions, giving rise to a dipole moment.” The expression of this sentence is not correct because it pertains to the situation of single-layer 2H MoS₂, whereas the value of BCD is 0.

Lines 41-43: “Therefore, topological materials with tilted Dirac or Weyl cones, which serve as sources of large BCD, have emerged as ideal candidates for realizing NLHE”. The wording of this sentence is also problematic. For instance, in the case of the type II Weyl semimetal WTe₂, the bulk BCD value is 0. Please refer to *Physical Review Letters* 130, 016301 (2023) for details.

Reviewer #3 (Remarks to the Author):

The manuscript by Bin Cheng et. al. reports the giant nonlinear Hall (NLH) at room temperature and also show the rectification effects in Te. Author shows the angular dependence of 2w Hall signal becoming maximum at 90 and 270 degrees angles with respect to c-axis. DC Hall signal, which is expected to be similar to 2w Hall data, is also recorded and shown that the dc signal follows the 2w signal. At a current of 50 μ A, the maximum Hall voltage is approximately 0.15 mV, a value that is notably increased to around 2.8 mV through the application of back gating. The central assertion of the manuscript revolves around the room temperature observation of this 2.8 mV Hall signal, albeit achieved through back gating.

The field focusing on nonlinear Hall effect and other phenomena governed by Berry phase and its curvature are growing interest in the scientific community from not only the fundamental but also from the application point of view. The manuscript under consideration, showcasing the giant nonlinear Hall effect in Te, possesses the potential to captivate readership and contribute significantly to a reputable journal. However, I have the following concern on the novelty and I cannot recommend this manuscript for the publication in Nature Communication.

My main concern pertain to the central claim of this manuscript, which is observing 2.8 mV NLH-voltage. First of all this is achieved by the application of back gating. The intrinsic signal is much smaller: around 0.15 mV. Also, the significance of the NLHE is not in getting just high voltage but to get high efficiency, which can be determined by obtaining the output power rather than just the voltage. Even better is to calculate the power efficiency; the ratio of output power with input power. As I can see, this material being semiconducting is highly resistive. The rough resistance I could estimate is in $\sim 10^4$ ohm order, therefore the voltage drop will automatically be high. This is nothing to do with large Berry curvature dipole or large extrinsic scatterings in Te. If I calculate the generated output power (V^2/R) then it will come out to be much smaller.

Apart from novelty concern, I have other comments mentioned below.

1. Can the author explicitly talk about which symmetries are broken on the surface and what is the origin behind the breaking of symmetry. Also, how significantly this symmetry is broken in Te because the breaking might be negligible on the surface.
2. What about the inversion symmetry in this material?
3. Request for evidence supporting the attribution of the NLH signal to the material's surface.
4. When the author says that the non-zero 2w signal in longitudinal direction indicates the existence of extrinsic scattering then in WTe₂ (Nature Material) and TaIrTe₄ (Nature Nanotechnology) where there is no longitudinal signal (only Hall voltages were observed) then following the author argument there should not be any contribution from extrinsic scatterings. However, a finite contribution from extrinsic scatterings was observed in both WTe₂ (Nature Material) and TaIrTe₄ (Nature Nanotechnology)?
5. I am not able to understand how did the author reach to the conclusion that since the Ar ion etched samples show opposite NLH signal therefore this effect is surface governed. Infact, it is also possible that initially the signal is coming from some other reasons and once the device is etched, it creates nonuniformity in the device causing nonlinear signal.
6. Author should perform 2w measurements on several Thicknesses of flakes without etching or damaging the surface.

7. I would like to see the results of higher order (3w, 4w etc) harmonic signal of Hall voltage.
8. What is the reason behind increasing the Hall signal in the presence of back gating and why it is asymmetric for opposite polarity of back gate?
9. In supplementary figure 3b, Author has shown the two terminal IV curves of their device. I would like to see the IV curves upto current of around 60-70 μ A or higher?
10. Supp Fig. 2a shows the semiconducting nature of their devices. Can the author comment on why the IV curves of semiconducting devices are linear?

Addressing these concerns and providing additional insights will strengthen the manuscript and enhance its suitability for publication in a prestigious journal.

+++++

Responses to reviewers' reports
(Manuscript NCOMMS-23-51855-T by Bin Cheng, et al.)

=====

Detailed responses to reviewer #1:

General Comments: *The manuscripts by Cheng et al. has demonstrated a strong room-temperature nonlinear Hall effect, and its rectification ability of microwave signal. I think this is an important work both in fundamental physics and novel electronic applications. It demonstrates the universality of geometric effects of electrons in solids. The wireless rectification effects could also enable potential useful applications.*

The authors have done a careful job in analyzing the nonlinear Hall effect, including studying the current, angle and fermi energy dependence. The data sufficiently supports their conclusion.

After all, I would like to recommend the papers for publication.

Reply: We thank the reviewer for recommending publication in *Nature Communications*. As shown in the response below, we have addressed the comment raised by the reviewer, and have revised the manuscript accordingly.

Comment 1: *I only have one minor question about the angular dependence of the nonlinear Hall effects. I wonder if the authors have etched the devices to make the current flow along the targeted directions. it looks like the device doesn't have a rotational-symmetric shape. I understand it may not affect the observation, but have the authors done some finite-element simulation to support their observation if they haven't etched the device?*

Reply: In the original manuscript, measurements on the angular-dependent nonlinear Hall effects (NLHE) were carried out on the as-grown trapezoidal Te flakes, which indeed don't have a rotational-symmetric shape. Similar measurement configurations actually have been employed in various studies on NLHE (*Nature* 621, 487 (2023); *Science* 381, 181 (2023); *Phys. Rev. Lett.* 130, 016301 (2023); *Natl. Sci. Rev.* nwad103 (2023)), with the experimental data consistent with the symmetry analysis.

Motivated by this comment, we have carried out new experiments on etched Te flakes possessing rotational-symmetric shapes, with the typical results shown in Fig. R1. A well-defined quadratic dependence on the applied current I^{ω} is evident in all of the measured nonlinear Hall voltages $V_{xy}^{2\omega}$ at various θ (Fig. R1a). By plotting the obtained $V_{xy}^{2\omega}$ at $I^{\omega} = 51 \mu\text{A}$ as a function of θ , a one-fold angular dependence is seen

(Fig. R1b). The maximum is achieved at θ around 90° and 270° , where the current is parallel to the a -axis. Such an angular dependency is identical to the results for the as-grown Te samples without rotational-symmetric shape (see Fig. 1d and the original Supplementary Fig. 5d), and can also be well fitted by the same Equation that is derived from symmetry analysis. These newly-obtained data clearly demonstrate that the sample shape has a negligible effect on the angular-dependent observations, and thus will not affect relevant claims in our manuscript.

We thank the reviewer for this constructive comment, and have included Fig. R1 as new Supplementary Fig. 5 in the revised manuscript. Related discussions have also been added in the main text (the 1st paragraph on page 4).

Figure R1 | **a**, Second-harmonic Hall voltage $V_{xy}^{2\omega}$ as a function of input current (I^ω) at different angles for a newly-fabricated sample with rotational-symmetric shape. θ is the angle between the c -axis and the current direction. The solid lines are the quadratic fitting results. Inset is the optical image of the device. **b**, Angular dependence of $V_{xy}^{2\omega}$ for $I^\omega = 51 \mu\text{A}$ and corresponding fitting result by using Equation (S14) in the revised Supplementary Information.

Detailed responses to reviewer #2:

General Comments: *In the manuscript, Cheng and co-authors present an impressive study of giant room temperature second-order Hall nonlinearity in 2D Te thin flakes. As for application, the tunability of nonlinear response and the wireless rectification effects in the GHz frequency band are clearly demonstrated. Although the physics origins related to quantum metric and Berry curvature dipole are fascinating, the simultaneous observation of longitudinal and Hall nonlinear voltages in this work points towards the extrinsic scattering mechanism. The control experiments and scaling fitting verify the skew-scattering origins. This work provides an attractive candidate in nonlinear Hall family and offers an essential step to the practical application. Therefore, I recommend this paper for publication in Nature Communications. Several questions for further improvement are listed as follows:*

Reply: We thank the reviewer for recommending publication in *Nature Communications*. As shown in the point-by-point responses below, we have addressed all the comments and suggestions raised by the reviewer, and have revised the manuscript accordingly. We appreciate all the reviewer's comments, which have resulted in a substantially improved paper with enhanced validity.

Comment 1: *In the manuscript, the crystal structure and the principal axis of Te are clearly shown in Fig.1. The b-axis is not perpendicular to the a-axis and also the a-c plane. Indeed, it reflects directly on the symmetry analysis in Supplementary Note 1. Specifically, the y-axis is not perpendicular to the x-z plane as set by the authors in Line 28 to Line 30 in Supplementary Information. Thus, the symmetry analysis and discussion in Supplementary Note 1 should be more complicated in principle. I wonder if the authors have taken this crucial point in consideration.*

Reply: We thank the reviewer for pointing out this issue. In the analysis presented in Supplementary Note 1, our focus was on the nonlinear Hall signal generated within the ac plane by applying an in-plane electric field, while any non-coplanar electric field was viewed as zero. Consequently, the orientation of the b-axis, whether perpendicular to the ac plane or not, doesn't impact the validity of the as-deduced equations. For the symmetry analysis, a general Cartesian coordinate ($x \perp y \perp z$) was employed, wherein the x and z axes align with the a and c axes of Te, respectively. The y-axis, while perpendicular to the ac plane, does not necessarily coincide with the b axis. Relevant statement in Lines 28-30 of the original Supplementary Note 1 is not accurate as the reviewer pointed out, and has been revised to avoid any potential misunderstandings.

Comment 2: *It would be better if the characterization of the as-grown Te samples by high-resolution transmission electron microscope and Raman spectrum can be added into the manuscript or supplementary information. Specially, the polarized Raman spectra are needed to directly certify the angular-dependent transport results.*

Reply: As was mentioned in the Method section, both transmission electron microscope (TEM) and Raman spectrum measurements were conducted to check the quality of the as-grown Te flakes, while the results were not shown in the original manuscript. Here we would like to present the typical data, which have been included in the revised Supplementary Information following the reviewer's suggestion (new Supplementary Fig. 1).

Figure R2 shows the typical high-resolution TEM image and selective area electron diffraction (SAED) pattern of the exposed surface of Te flake, from which the high crystalline quality of the sample is clearly seen. The surface is identified as the ac plane,

with lattice parameters determined as $a = 4.60 \text{ \AA}$, $c = 6.04 \text{ \AA}$.

The typical data for the polarized Raman characterizations are shown in Fig. R3a. There are three peaks visible, corresponding to E_1 -TO, A_1 and E_2 modes, at 91 cm^{-1} , 120 cm^{-1} and 139 cm^{-1} , respectively. Figs. R3b-d further show the angular-dependent peak intensities of the three modes that were extracted from Fig. R3a. Notably, a two-lobe shape for E_1 -TO and A_1 modes and a four-lobe shape for E_2 mode are observed, all of which can be effectively described by the corresponding Raman tensors (*Phys. Rev. B* 4, 356 (1971)). The maximum intensity of E_1 -TO, A_1 and E_2 modes occurs respectively at $\theta = n\pi/2$, $n\pi/2$ and $n\pi/4$ ($n = 1, 3, 5 \dots$), which is in good agreement with previous Raman studies of Te [*Nat. Electron.* 1, 228 (2018); *Nat. Commun.* 11, 2308 (2020)]. This result further confirms the anisotropic crystallographic orientation of the as-grown trapezoidal-shaped Te flake, where the long edge of the flake sample represents the c -axis, i.e., the direction of the 1D atomic chains. Just because of the unique 1D atomic chain structure, the measured resistance within the ac plane reaches its minimum when the current is applied along the direction of atomic chains (c -axis), as typically shown in Fig. 1b in the main text.

Figure R2 | **a**, High-resolution TEM and **b**, SAED pattern of the as-grown Te thin flake.

Figure R3 | **a**, Angle-resolved polarized Raman spectra for the as-grown Te flake. θ is defined as the angle between the polarization direction of laser and the long edge of the Te sample with trapezoidal shape (see the inset). **b-d**, Polar plots of peak intensity for A_1 , E_1 -TO, and E_2 modes, respectively. The red dots are experimental data and the blue dashed lines are corresponding fitting curves. The fitting equations for these three modes are $I_{A_1} = |a\sin^2\theta + b\cos^2\theta|^2$, $I_{E_1\text{-TO}} = |c\sin^2\theta|^2$ and $I_{E_2} = |2d\sin\theta\cos\theta|^2$, where a , b , c , and d are Raman tensor elements (*Phys. Rev. B* 4, 356 (1971)).

Comment 3: *The two-fold rotation C_2 about the a -axis is supposed to be broken at the surface but the three-fold rotation C_3 about the c -axis is preserved. However, the three rows of Te atoms in the 1D triple helix chain at the surface experience different atomic environments. The C_3 symmetry could be broken at the surface in principle.*

Reply: We thank the reviewer for this insightful comment. The claim for the reduction of crystalline symmetry at the surface is based on the fact that the inherent D_3 point group of bulk Te does not allow the existence of nonlinear Hall signal along the principal axes, thus contradicting with our observations shown in Fig. 1 and the original Supplementary Fig. 5. Instead, when considering that C_2 symmetry is broken at the surface, nonzero Hall signal will emerge, and the obtained angular-dependent nonlinear Hall data can be well described by the equations deduced from corresponding symmetry analysis.

We totally agree with the reviewer that the C_3 symmetry might also be broken at the

surface in principle. However, if one only considers the breaking of C_3 symmetry, the residual C_2 symmetry also prohibits the emergence of nonlinear Hall response along the c-axis (as detailed in the revised Supplementary Note 1). Alternatively, if assuming that **C_3 and C_2 symmetries are both broken**, the point group is reduced to C_1 . In this case, the corresponding equation for the angular dependent 2ω Hall signal is expressed as follows (also refer to the revised Supplementary Note 1):

$$E_{\perp}^{(2)} = E_x^{(2)} \cos \theta + E_z^{(2)} \sin \theta = \rho_a^3 j^2 [\chi_{zxx} r \sin^3 \theta + \chi_{zzz} r^2 \cos^3 \theta + (\chi_{xxx} - 2\chi_{zzx} r^2) \sin^2 \theta \cos \theta + (\chi_{zzz} r^3 - 2\chi_{xxz} r) \sin \theta \cos^2 \theta] \quad (R1)$$

As shown in Fig. R4, this equation fits the experimental data better than the equation that assumes only C_2 symmetry breaking.

Therefore, to account for the features of the nonlinear Hall response in Te, reduction of symmetry at the surface is essential. The breaking of C_2 symmetry is the minimal requirement, while concurrently violating C_2 and C_3 symmetries provides an improved fit to the experimental results. Accordingly, we have revised relevant discussions on the symmetry analysis by further considering the case that C_2 and C_3 are both broken (the 1st paragraph on page 8 in the revised manuscript), with more details incorporated in the new Supplementary Note 1.

Figure R4 | Fitting results for the angular dependent data by using the as-deduced equations for two different cases of surface symmetry breaking. C_3 Fitting (grey line): only C_2 symmetry is broken, and the point group is reduced to C_3 . C_1 Fitting (pink line): both C_3 and C_2 symmetries are broken, and the point group is reduced to C_1 . Here, the experimental data are the same as those shown in Fig. 1d.

Comment 4: *The results of angular-dependent measurements of the second-order Hall voltage $V_{xy}^{2\omega}$ have been fitted in the Fig. 1d using the equation in the Supplementary Information. It would be better if the authors add the angular-dependent fitting results of the observed longitudinal second-order voltage $V_{xx}^{2\omega}$ measured as shown in Fig. 1e in the manuscript or supplementary information. Besides, if the analysis in Supplementary Note 1 is correct, the fitting coefficients of the Hall results and*

longitudinal results could be similar. Moreover, the fitting results of $V_{xx}^{2\omega}$ could help to eliminate the thermoelectric effect more persuasively. Because the Joule heating mainly contributes to the longitudinal transport.

Reply: In the original manuscript, the angular-dependent second-order Hall data were fitted via the as-deduced equation by considering the surface C_2 symmetry breaking. However, as illustrated in the above reply, the simultaneous breaking of C_2 and C_3 symmetries yields a better fitting result. Here, we would like to respond to this comment under such a consideration as well.

When both C_3 and C_2 symmetries are broken, the equations for the as-generated transverse (Hall) and longitudinal components of the second-order electric field can be written as:

$$E_{\perp}^{(2)} = E_x^{(2)} \cos \theta + E_z^{(2)} \sin \theta = \rho_a^3 j^2 [\chi_{zxx} r \sin^3 \theta + \chi_{zzz} r^2 \cos^3 \theta + (\chi_{xxx} - 2\chi_{zzx} r^2) \sin^2 \theta \cos \theta + (\chi_{zzz} r^3 - 2\chi_{xxz} r) \sin \theta \cos^2 \theta] \quad (R1)$$

and

$$E_{//}^{(2)} = E_z^{(2)} \cos \theta - E_x^{(2)} \sin \theta = \rho_a^3 j^2 [-\chi_{xxx} \sin^3 \theta + \chi_{zzz} r^3 \cos^3 \theta + r(2\chi_{xxz} + \chi_{zzx}) \sin^2 \theta \cos \theta - r^2(\chi_{zzz} + 2\chi_{zzx}) \sin \theta \cos^2 \theta] \quad (R2)$$

These two equations do share the same set of parameters, just as the reviewer pointed out.

For device #D1 presented in Fig. 1, the angular dependent data of $V_{xx}^{2\omega}$ had been collected simultaneously with the Hall component $V_{xy}^{2\omega}$, as shown in Fig. R5. The corresponding fittings for the $V_{xy}^{2\omega}$ and $V_{xx}^{2\omega}$ data should be performed by using Equations (R1) and (R2), respectively. Nevertheless, by fitting $V_{xy}^{2\omega}$ or $V_{xx}^{2\omega}$ individually, we can obtain not each response coefficient but their certain combination. Since $V_{xx}^{2\omega}$ and $V_{xy}^{2\omega}$ depend on the response coefficients differently, we cannot directly compare the fitting results. Instead, we can combine them to yield a single set of parameters that can well capture both $V_{xx}^{2\omega}$ and $V_{xy}^{2\omega}$. This approach has been commonly used in data fitting when two equations share same parameters (*Phys. Rev. Lett.* 115, 027006 (2015); *Phys. Rev. B* 75, 115114 (2007)). Here we have adopted such fitting approach, and the corresponding results are shown in Fig. R5 (solid lines). It is clear that both the longitudinal and Hall data can be well described by using the same set of parameters.

Regarding the possible thermoelectric effect on the measured second-harmonic voltage output, we had carefully ruled it out in the original Supplementary Note 4. Here, the fitting result for the angular dependent $V_{xx}^{2\omega}$ data could add further evidence, just as

the reviewer commented. Typically, the anisotropic $V_{xx}^{2\omega}$, which is closely linked to the crystalline symmetry of Te, cannot be explained by the thermoelectric effect, which is normally arbitrary within the sample plane.

We appreciate this insightful comment raised by the reviewer. In the revised manuscript, we have included Fig. R5 as the new Supplementary Fig. 9, with relevant discussions added in the new Supplementary Notes 1 and 4.

Figure R5 | Second-harmonic Hall (upper panel) and longitudinal (lower panel) voltage as a function of θ . The solid lines are the fitting curves, which were obtained by combining Equations (R1) and (R2) into a single group and then fitting the data for $V_{xy}^{2\omega}$ and $V_{xx}^{2\omega}$ simultaneously.

Comment 5: *As shown in Supplementary Fig.1a, the first-harmonic measurements show the Hall voltage is not zero. I think the reason should be clarified in the manuscript or supplementary information. If it originates from the unavoidable misalignment, it may also be another origin of the observed nonlinear Hall voltage or longitudinal voltage.*

Reply: The observed first-harmonic Hall voltage does exhibit a finite value, as the reviewer pointed out. The unavoidable misalignment of the electrodes should be the source of such a non-zero Hall signal under zero magnetic field. Nevertheless, the corresponding amplitude of such a Hall signal remains very small when compared to the longitudinal one, for example, only 4% for the data presented in the original Supplementary Fig.1a. The contribution of longitudinal-signal-induced mixing to the observed NLHE can thus be reasonably neglected, according that the amplitude of the measured second-harmonic Hall voltage is comparable to that of the longitudinal one (see Fig. 1e).

Motivated by this comment, we have added relevant discussions in Supplementary Note

4 in the revised manuscript.

Comment 6: *The two results from up and down channel of Supplementary Fig. 1b are different. The 2ω and DC results are more identical in the down channel but the results in the up channel are not.*

Reply: The upper and lower panels of the original Supplementary Fig. 1b represent two measurement configurations with opposite current directions. There are two features related to the reviewer's comment: **(I)**. there is a slight difference between the 2ω and DC Hall voltages, and **(II)** such divergence also shows some asymmetry for opposite current directions. Actually, both features have been observed in other materials exhibiting NLHE, such as few-layer WTe₂ shown in Fig. R6 (see Fig. 2c in **Ref. Nat. Mater.** 18, 324 (2019)).

As for feature **I**, one possible reason is that the rectification measurement for the DC output is more susceptible to external disturbances, in comparison to the collection of 2ω signal by using lock-in technique. Regarding feature **II**, we are unable to provide a clear explanation at this time. It could be caused by some asymmetric factors, such as sample shape, electrode contact, and even the background signal of the voltmeter.

[REDACTED]

Figure R6 | Second-harmonic and DC Hall voltage of WTe₂. The black dots are obtained with the directions of the current and Hall probes inversed simultaneously, as compared with the measurement configuration for the red dots (*Nat. Mater.* 18, 324 (2019)).

Comment 7: *The thickness-dependent measurements are lacked for proof of the surface effect discussed in the Line 184 to Line 196 in manuscript.*

Reply: As demonstrated in the original manuscript, the claim that surface effect accounts for the observed NLHE is strongly supported by both theoretical fitting for the angular-dependent data and experimental studies on etched samples. Here, motivated by the reviewer's comment, we have fabricated additional devices and measured nonlinear transport by using as-grown Te flakes with different thicknesses (without etching). Figure R7 shows the measured second-harmonic Hall voltages for these devices, all of which clearly exhibit a quadratic dependence on the applied AC

current. As the thickness increases, the magnitude decreases noticeably. For instance, the nonlinear Hall voltage for Te flake with a thickness of 24 nm is 0.21 mV (Fig. R7a). While for the 131 nm thick sample, this value drops to about 0.003 mV, which is nearly two orders of magnitude smaller (Fig. R7c). This result provides unequivocal additional evidence supporting the proposed surface effect.

We appreciate the reviewer's constructive comment. In the revised manuscript, we have included Fig. R7 as the new Supplementary Fig. 10 and have revised related statement in the main text (the 2nd paragraph on page 8), with more details incorporated in the new Supplementary Note 5.

Figure R7 | **a**, Second-harmonic Hall voltage as a function of the AC current for devices H#3 (thickness: 24 nm, as already shown in the original Supplementary Fig. 4b). **b**, **c**, The measured data for two newly-fabricated devices with larger thickness (75 nm and 131 nm). The red lines are corresponding $(I^\omega)^2$ fitting curves. Insets in **a-c**: optical images of the devices.

Comment 8: *The reported results (Q. Fu, et al. Berry Curvature Dipole Induced Giant Mid-Infrared Second-Harmonic Generation in 2D Weyl Semiconductor. Adv. Mater. 2306330 (2023)) show the Berry curvature dipole induced nonlinear optical physics in 2D Te. Due to the broken inversion symmetry of 2D Te, the Berry curvature dipole exists along particular directions. Could such non zero Berry curvature dipole affect the measured nonlinear transport phenomena?*

Reply: We thank the reviewer for this insightful comment. In reference *Adv. Mater. 2306330 (2023)*, the authors have calculated the Berry curvature dipole (BCD) of Te, and claimed that the observed second-harmonic generation signal can be attributed to the presence of non-zero BCD. In fact, we had already conducted similar DFT calculations, aiming to assess the possible contribution from inherent BCD to the observed NLHE. Because of crystallographic symmetry constraints, the BCD for bulk Te that belongs to the D_3 point group, even if it is not zero, cannot contribute a nonlinear Hall signal along the principal axes (*Nat. Commun.* 10, 3047 (2019)). Figure R8 shows our computation results at the surface, from which the BCD component D_{xy} is clearly seen. Here, we solely offer the calculation for D_{xy} , since it is the only component that might contribute to the nonlinear Hall response along the z-axis (*Phys. Rev. Lett.* 115, 216806 (2015)).

Notably, for the intrinsic BCD mechanism under relaxation time approximation, the second-harmonic longitudinal response is prohibited [*Phys. Rev. Lett.* 115, 216806 (2015); *Nature* 565, 337 (2019)]. However, in our experiments, we have detected strong second-harmonic signals in the longitudinal direction, whose amplitudes were comparable with those of the Hall signals. This finding strongly implies that, in comparison to the intrinsic BCD origin, scattering-related extrinsic mechanism dominates, which is further supported by the scaling law analysis detailed in Supplementary Note 3.

Figure R8 | Calculated Berry curvature dipole D_{xy} within the k_x - k_z plane. Here, Cartesian coordinates (x, y, z) are employed, among which the x, z are selected along Te crystal’s principal axes, i.e., a- and c-axes, and y is perpendicular to the xz plane. The Fermi level is set below the top of valence band during the calculation, according to the intrinsic hole-doping property of the Te samples.

Comment 9: Lines 40-41: “That is, inversion-symmetry breaking may segregate the positive and negative Berry curvatures in different momentum regions, giving rise to a dipole moment.” The expression of this sentence is not correct because it pertains to the situation of single-layer 2H MoS₂, whereas the value of BCD is 0.

Lines 41-43: “Therefore, topological materials with tilted Dirac or Weyl cones, which serve as sources of large BCD, have emerged as ideal candidates for realizing NLHE”. The wording of this sentence is also problematic. For instance, in the case of the type II Weyl semimetal WTe₂, the bulk BCD value is 0. Please refer to *Physical Review Letters* 130, 016301 (2023) for details.

Reply: We appreciate the reviewer’s thorough reading and pointing out our carelessness. In the case of single-layer MoS₂, the inversion symmetry is spontaneously broken and significant Berry curvature develops close to the valleys. However, the BCD must be zero due to the presence of C_{3v} symmetry (*Phys. Rev. B* 103, 235151 (2021)). Regarding the type-II Weyl semimetal WTe₂, the combination of mirror symmetry M_a and the glide mirror symmetry M_b also forbids the existence of bulk BCD (*Phys. Rev. Lett.* 130,

016301 (2023)). The two examples raised by the reviewer clearly demonstrate the essential role that the crystalline symmetry constraint plays in the formation of non-zero BCD. This issue, however, was ignored in our original statements.

To avoid further misunderstandings, we have revised the statements in Lines 40-43 according to this comment, as follows: “*That is, inversion-symmetry breaking may segregate the positive and negative Berry curvatures in different momentum regions, giving rise to a dipole moment when the crystalline symmetry allows. Therefore, topological materials with tilted Dirac or Weyl cones, which serve as sources of large BCD, have emerged as ideal candidates for realizing NLHE.*”

Detailed responses to reviewer #3:

General Comments: *The manuscript by Bin Cheng et. al. reports the giant nonlinear Hall (NLH) at room temperature and also show the rectification effects in Te. Author shows the angular dependence of 2w Hall signal becoming maximum at 90 and 270 degrees angles with respect to c-axis. DC Hall signal, which is expected to be similar to 2w Hall data, is also recorded and shown that the dc signal follows the 2w signal. At a current of 50 μA , the maximum Hall voltage is approximately 0.15 mV, a value that is notably increased to around 2.8 mV through the application of back gating. The central assertion of the manuscript revolves around the room temperature observation of this 2.8 mV Hall signal, albeit achieved through back gating.*

The field focusing on nonlinear Hall effect and other phenomena governed by Berry phase and its curvature are growing interest in the scientific community from not only the fundamental but also from the application point of view. The manuscript under consideration, showcasing the giant nonlinear Hall effect in Te, possesses the potential to captivate readership and contribute significantly to a reputable journal. However, I have the following concern on the novelty and I cannot recommend this manuscript for the publication in Nature Communication.

Reply: We thank the reviewer for recognizing the quality and potential impacts of our work. However, the reviewer did express some reservations about the novelty of our work, particularly with regard to the superior NLHE performance of Te (see Comment 1 below). In addition, a number of comments on the analysis of some experimental findings were also proposed. As shown in the point-by-point reply below, we have addressed all the reviewer’s comments with more refined discussions and newly-obtained experimental data, and have revised the manuscript accordingly. We appreciate all the reviewer’s questions and suggestions, which have led us to

substantially improve the manuscript with enhanced validity.

Comment 1: *My main concern pertains to the central claim of this manuscript, which is observing 2.8 mV NLH-voltage. First of all this is achieved by the application of back gating. The intrinsic signal is much smaller: around 0.15 mV. Also, the significance of the NLHE is not in getting just high voltage but to get high efficiency, which can be determined by obtaining the output power rather than just the voltage. Even better is to calculate the power efficiency; the ratio of output power with input power. As I can see, this material being semiconducting is highly resistive. The rough resistance I could estimate is in $\sim 10^4$ ohm order; therefore the voltage drop will automatically be high. This is nothing to do with large Berry curvature dipole or large extrinsic scatterings in Te. If I calculate the generated output power (V^2/R) then it will come out to be much smaller. Apart from novelty concern, I have other comments mentioned below.*

Reply: We respond to this comment in two aspects:

- (1) The output voltage has been widely adopted as a key indicator for evaluating the strength of nonlinear Hall effect (*Nature* 565, 337 (2019); *Nat. Rev. Phys.* 3, 744 (2021)). Nevertheless, we agree with the reviewer that the power efficiency might be a better metric, since it is dimensionless and could eliminate the impact of sample resistance. Following the reviewer's suggestion, we have estimated the power efficiency for Te devices using the following expression:

$$P_{\text{out}}/P_{\text{in}} = \frac{(V_{\text{xy}}^{2\omega})^2}{R_{\text{out}}} / \frac{(V_{\text{xx}}^{\omega})^2}{R_{\text{in}}} = \frac{R_{\text{in}}}{R_{\text{out}}} \left(\frac{V_{\text{xy}}^{2\omega}}{V_{\text{xx}}^{\omega}} \right)^2 \quad (\text{R3})$$

where R_{in} and R_{out} are the load resistance for the input (longitudinal direction) and the output (Hall direction), respectively.

Figure R9a shows the gate-voltage dependent power efficiency for the same Te device originally presented in Fig. 2. The power efficiency increases monotonically as the V_{BG} increases, suggesting that gate-tuning has an impact beyond merely adjusting the sample resistance. By further comparing these values with those of other reported systems exhibiting NLHE (see Fig. R9b), it is evident that our Te device holds the highest power efficiency record at room temperature (RT), with a value that is an order of magnitude higher than that of BaMnSb₂.

- (2) The reviewer also expressed his/her concern about the use of gate tuning, by arguing that the intrinsic signal is not large enough, and the achievement of giant nonlinear Hall output relies on the gate tuning. Here, we would like to emphasize that the accessibility of gate tuning is actually a significant advantage for semiconductor systems like Te. This is also the primary driving force behind our investigation of

potential RT NLHE in a semiconductor system.

Despite the rapid progress in NLHE study, previous reports of **RT NLHE** have been limited to only two semimetal systems, namely BaMnSb₂ and TaIrTe₄. The inherent lack of tunability in these systems restricts further manipulation of the NLHE strength and, consequently, related device applications. Electrostatic gating is a well-established powerful approach for modulating the transport performance of a semiconductor, and therefore lays the foundation for practical applications. As evidenced by both the Hall voltage output (Fig. 2b) and the power efficiency data (Fig. R9a), the strength of NLHE in Te can be readily enhanced by several orders of magnitude. The realization of giant and highly-tunable NLHE makes Te an ideal system for actualizing applications in nonlinear electronic devices, in addition to its scientific significance in the exploration of nonlinear transport. The primary message we wish to convey in our work is the identification of the first semiconductor demonstrating RT NLHE.

Motivated by this comment, we have included Fig. R9 into the revised manuscript as new Supplementary Fig. 6, with related discussions added in the main text (the 1st paragraph on page 6).

Figure R9 | **a**, Power efficiency (P_{out}/P_{in}) as a function of V_{BG} obtained at 300 K for Te device #D1 (same as the one presented in Fig. 2). **b**, Comparison of the NLHE between different systems by using P_{out}/P_{in} as the indicator. Here the values of the other systems are calculated using the data of $V_{xy}^{2\omega}$ and V_{xx}^{ω} from corresponding references. Since the precise values of R_{in} and R_{out} were usually not provided in the majority of these references, the ratio of R_{in}/R_{out} was assumed to be 1 during the calculations. Such a simplified process is reasonable, since the reported in-plane resistance anisotropy for these systems are always small, e.g., around 1.14 for BaMnSb₂ (*Nat. Commun.* 14, 364 (2023)).

Refs. 1-8: *Nature* 565, 337 (2019); *Nat. Nanotechnol.* 16, 421 (2021); *Phys. Rev. Lett.* 129, 186801 (2022); *Nat. Mater.* 18, 324 (2019); *Chin. Phys. Lett.* 38, 017301 (2021); *Nat. Electron.* 4, 116 (2021); *Nat. Commun.* 12, 698 (2021); *Nat. Commun.* 14, 364 (2023).

Comment 2: *Can the author explicitly talk about which symmetries are broken on the surface and what is the origin behind the breaking of symmetry. Also, how significantly*

this symmetry is broken in Te because the breaking might be negligible on the surface.

Reply: We thank the reviewer for this insightful comment. We respond to this comment in three aspects:

- (1) *Which symmetries are broken?* In the original manuscript, C_2 symmetry on the surface was supposed to be broken. For bulk Te, the inherent D_3 point group does not allow the experimentally observed nonlinear Hall signal along the principal axes, which motivated us to consider the potential of symmetry degradation on the sample surface. When considering that C_2 symmetry is broken on the surface, nonzero Hall signal will emerge, and the obtained angular-dependent nonlinear Hall data can be well described by the equation deduced from corresponding symmetry analysis. Nevertheless, as detailed in the reply to Reviewer #2's Comment 3, the simultaneous breaking of C_2 and C_3 symmetries yields a better fitting result. It is therefore more reasonable to consider that the C_2 and C_3 symmetries are both broken on the surface.
- (2) *The origin behind the breaking of symmetry.* Surface symmetry breaking is frequently observed in low-dimensional systems, which could have various sources, including strain (*Nat. Commun.* 3, 1189 (2012)), interfacial coupling (*Phys. Rev. Lett.* 110, 076801 (2013)), and surface termination (*Science* 344, 488 (2014)). Previous reports have taken the surface effect into account, with the goal of explaining NLHE in materials whose pristine symmetry prevents the emergence of a second-harmonic Hall signal (*Phys. Rev. B* 98, 121109(R) (2022); *Nat. Mater.* 18, 324 (2019); *Nat. Nanotechnol.* 16, 421 (2021)). For Te thin flakes, the atoms in the 1D helical chain experience different atomic environment on the surfaces, and the substrate may further induce additional tension onto the bottom surface. Both factors could lead to the breaking of the surface symmetry.
- (3) *How significantly this symmetry is broken?* From point (1), the surface symmetry breaking is essential for describing the features of nonlinear Hall data. Such surface effect is further supported by our findings on etched samples, where slight alternation in the surface condition leads to an evident change of nonlinear Hall signal. Moreover, the newly-conducted measurements on the as-grown samples with different thicknesses (without etching) could provide unequivocal additional evidence for the surface effect (see the reply to the reviewer's comment 7). By combing these factors, the breaking of surface symmetry plays a key role in inducing the giant NLHE in Te thin flakes. Nevertheless, quantitative evaluation of the degree of symmetry breaking requires further research efforts.

Motivated by this comment, we have added more in-depth discussions on the symmetry analysis in the revised manuscript (the 1st paragraph on page 8), with more details incorporated in the new Supplementary Note 1.

Comment 3: *What about the inversion symmetry in this material?*

Reply: Te has a highly anisotropic crystal structure that belongs to the D_3 point group, as schematically shown in Fig. R10. The atoms form helical chains along the c-axis by covalent bonding, while neighboring chains are bonded into a hexagonal lattice in the ab plane via van der Waals forces. There are only two types of symmetric operations in Te crystal, i.e., a three-fold rotation C_3 about the c-axis and two-fold rotation C_2 within the ab plane. **The inversion symmetry is already broken in such a material.**

Motivated by this comment, we have added Fig. R10 as new Supplementary Fig. 1, as well as more discussions about the crystalline symmetry of bulk Te in the new Supplementary Note 1.

Figure R10 | **a**, Schematic of Te crystal structure. **b**, Top and perspective views of the crystal structure, in which the crystalline symmetries are marked.

Comment 4: *Request for evidence supporting the attribution of the NLH signal to the material's surface.*

Reply: In our original manuscript, the claim of surface-dominated NLHE is mainly supported by two facts: **1**). the angular-dependent NLH data could be well described by the as-derived equations after taking surface symmetry breaking into consideration, and **2**). Experiments on etched samples clearly demonstrated that slight change to the surface condition could have an evident impact on the NLH behavior.

As motivated by the reviewer's comment 7, we have carried out further measurements on samples with different thicknesses (without etching). For Te flakes with larger thickness, the resultant NLH signals become significantly weaker, which provides additional evidence for the surface effect.

Comment 5: *When the author says that the non-zero 2ω signal in longitudinal direction indicates the existence of extrinsic scattering then in WTe₂ (Nature Material) and TaIrTe₄ (Nature Nanotechnology) where there is no longitudinal signal (only Hall voltages were observed) then following the author argument there should not be any contribution from extrinsic scatterings. However, a finite contribution from extrinsic scatterings was observed in both WTe₂ (Nature Material) and TaIrTe₄ (Nature Nanotechnology)?*

Reply: The key of this issue lies in that the observation of a finite longitudinal 2ω signal is a sufficient, but not a necessary condition for the existence of extrinsic scatterings, since the latter is also subject to the point-group symmetry. For the two systems raised by the reviewer (WTe₂ and TaIrTe₄), the estimation of finite contribution from extrinsic scattering was derived through scaling law analysis. Experimentally, the corresponding longitudinal signal is indeed negligible (*Nat. Mater.* 18, 324 (2019); *Nat. Nanotechnol.* 16, 421 (2021)), due to the crystalline symmetry constraint. Below we present a detailed explanation by taking WTe₂ as an example.

WTe₂ holds a non-centrosymmetric space group Pmn2₁. The combination of mirror symmetry M_a and the glide mirror symmetry M_b actually forbids the appearance of bulk BCD (*Phys. Rev. Lett.* 130, 016301 (2023)). During the measurements reported in **Ref. Nature** 565, 337 (2019), the AC current (ω) was applied along the a-axis within the ab plane, resulting in a 2ω Hall signal along the b-axis. However, the longitudinal signal measured along the a-axis was negligible. To explain these observations, glide mirror symmetry M_b was thought to be broken at the surface, and the space group symmetry is further reduced to Pm. Considering a general expression for the as-generated nonlinear Hall current: $\mathbf{J}_x = \chi_{xyz} \mathbf{E}_y \mathbf{E}_z$, the corresponding nonlinear susceptibility of Pm is described as (*Nat. Commun.* 12, 5038 (2021)):

$$\chi = \begin{pmatrix} 0 & \chi_{aab} & 0 & \chi_{aac} & 0 & 0 \\ \chi_{baa} & 0 & \chi_{bbb} & 0 & \chi_{bbc} & \chi_{bcc} \\ \chi_{caa} & 0 & \chi_{cbb} & 0 & \chi_{ccb} & \chi_{ccc} \end{pmatrix},$$

where a , b , and c stand for the crystal's principal axes. Notably, a non-zero element χ_{baa} is present in the susceptibility matrix, implying the potential for achieving a finite 2ω Hall signal on the b-axis when an AC current is applied along the a-axis. However, the lack of element χ_{aaa} suggests that the corresponding longitudinal signal along the a-axis is zero. These results are consistent with the experimental data, confirming that the longitudinal 2ω signal is indeed forbidden by the crystalline symmetry.

Comment 6: *I am not able to understand how did the author reach to the conclusion that since the Ar ion etched samples show opposite NLH signal therefore this effect is*

surface governed. In fact, it is also possible that initially the signal is coming from some other reasons and once the device is etched, it creates nonuniformity in the device causing nonlinear signal.

Reply: Our motivation to explore the surface effect stems from the fact that the bulk crystalline symmetry of Te does not accommodate the observed angular-dependent nonlinear Hall data. When surface symmetry is taken into account, the as-derived equations from symmetry analysis could well describe the experimental results. In order to verify the surface effect, we performed nonlinear Hall measurements on the etched samples, in which etching only modifies the surface condition, while leaving the bulk intact. The pronounced alterations of the measured NLH behavior, especially the reversal of the sign, clearly demonstrate the high sensitivity of NLH response to the surface condition. The scaling law analysis of the NLH signal indicates that the observed sign reversal depends directly on the modification of contributions from different types of scattering (as detailed in Supplementary Notes 2 and 3), which arises from the etching effect on the surface.

We agree with the reviewer that etching-induced nonuniformity may affect the nonlinear transport. However, according to the atomic force microscope image shown in Fig. R11, etching just modifies surface roughness and reduces the thickness. No obvious spatial nonuniformity is observed for the surface of the etched regions, even in region V, which has the highest degree of etching.

Figure R11 | **a**, Atomic force microscope (AFM) image of the Hall bar device #H1 for the etching measurement. **b**, **c**, Enlarged AFM images of Region I (unetched) and V (etched), respectively. The average roughness for Region I and V is respectively 0.139 nm and 0.316 nm.

Comment 7: *Author should perform 2w measurements on several Thicknesses of flakes without etching or damaging the surface.*

Reply: Following the reviewer’s suggestion, we have performed new measurements on

the as-grown Te flakes (without etching) with different thicknesses, and the results were shown in Fig. R12. It is clear that all the measured second-harmonic Hall voltages clearly exhibit a quadratic dependence on the applied AC current. As the thickness increases, the magnitude decreases noticeably. For instance, the nonlinear Hall voltage for Te flake with a thickness of 24 nm is 0.21 mV (Fig. R12a). While for the 131 nm thick sample, this value drops to about 0.003 mV, which is nearly two orders of magnitude smaller (Fig. R12c). The newly-conducted experiments on the unetched samples provide unequivocal additional evidence supporting the proposed surface effect.

We appreciate the reviewer's insightful comment. In the revised manuscript, we have included Fig. R12 as the new Supplementary Fig. 10, and have revised related statements in in the main text (the 2nd paragraph on page 8), with more details incorporated in the new Supplementary Note 5.

Figure R12 | **a**, Second-harmonic Hall voltage as a function of the AC current for devices H#3 (thickness: 24 nm, as already shown in the original Supplementary Fig. 4b). **b, c**, The measured data for two newly-fabricated devices with larger thickness (75 nm and 131 nm). The red lines are corresponding $(I^\omega)^2$ fitting curves. Insets in **a-c**: optical images of the devices.

Comment 8: *I would like to see the results of higher order (3 ω , 4 ω etc) harmonic signal of Hall voltage.*

Reply: Following the reviewer's suggestion, we have measured the nonlinear Hall effect at higher orders (3 ω and 4 ω), and the typical results are shown in Fig. R13. Non-zero 3rd and 4th harmonic Hall signals are observed (Figs. R13b, c), and the measured voltage shows distinct cubic and quartic dependences on the applied current I^ω , respectively. In comparison to the 2 ω case (see Fig. R13a), the 3 ω and 4 ω signals are considerably smaller. We appreciate the reviewer's comment, which motivates us to explore higher-order nonlinear transport in Te in the upcoming period.

Figure R13 | a-c, 2nd, 3rd and 4th harmonic non-linear Hall voltages as functions of the applied AC current measured at 300 K and corresponding fitting curves.

Comment 9: *What is the reason behind increasing the Hall signal in the presence of back gating and why it is asymmetric for opposite polarity of back gate?*

Reply: We thank the reviewer for this insightful comment. Since our Te samples are intrinsically hole-doped, applying a positive back-gate voltage will shift the Fermi level towards the bandgap. The as-induced increase of resistivity is one reason for the increase of the 2nd harmonic Hall voltage under positive gate voltage as shown in Fig. 2. On the other hand, the strengths of scattering and Berry curvature dipole are normally diverging at the band edge (*Phys. Rev. Lett.* 115, 216806 (2015); *Nat. Commun.* 10, 3047 (2019)), e.g., the top of valence band for our Te samples. This could also contribute to the increase of NLHE strength when applying a positive gate voltage.

As for the asymmetric Hall signal obtained at opposite polarity of the back gate, this phenomenon is also attributed to the hole-doping property of the Te samples. With respect to the 0 V case, applying a positive (negative) gate voltage will shift the Fermi level towards (away from) the bandgap, and thus result in nonlinear Hall signals with opposite tendencies.

Comment 10: *In supplementary figure 3b, Author has shown the two terminal IV curves of their device. I would like to see the IV curves upto current of around 60-70μA or higher?*

Reply: Figure R14 shows the *I-V* curves within a wider range of source-drain voltage for device #D1 (same with the one presented in the original Supplementary Fig. 3b). Even when the measured current reaches up to ~100 μA, a well-defined linear dependence is maintained for arbitrary two electrodes, indicating the good Ohmic contact of the electrodes. We thank the reviewer's comment, and have replaced the original Supplementary Fig. 3b with Fig. R14b in the revised manuscript.

Figure R14. **a**, Optical image of the disc device #D1. **b**, Two-terminal DC I - V curves between arbitrary two electrodes.

Comment 11: *Supp Fig. 2a shows the semiconducting nature of their devices. Can the author comment on why the I - V curves of semiconducting devices are linear?*

Reply: Nonlinear I - V curves are the result of poor electrode-to-sample contact in semiconductor devices. Such an imperfect contact normally originates from the formation of a Schottky barrier due to the mismatch of work functions between the electrodes and the sample. In previous transport studies of Te, metals with high work functions (such as Pd, Pt, and Ni) were normally used as electrode material because the as-grown Te samples are typically hole-doped, and good Ohmic contact has been widely achieved (*Nano Lett.* 17, 3965 (2017); *Nano Lett.* 18, 5760 (2018); *Nat. Mater.* 21, 526 (2022)). Following these studies, Pd was adopted as electrode material in the present study and linear I - V curves were achieved.

REVIEWER COMMENTS

Reviewer #1 (Remarks to the Author):

The authors have addressed my comments in full, I recommend their publication in its current form.

Reviewer #2 (Remarks to the Author):

I would like to thank the authors for their additional experiments and discussions for the giant nonlinear longitudinal and transverse nonlinearity in tellurium. Most of my questions are clearly answered, and especially I appreciate the additional careful symmetry analysis for the discussions of the surface effect.

The following points are provided for the authors' consideration, aimed at further enhancing the quality and impact of this manuscript.

1. The angular dependence of nonlinear voltages at zero gate voltage is shown in the manuscript and Supplementary materials. However, the angular-dependent measurements with nonzero gate voltages are necessary to exclude the gate-induced nonuniformity and complete the experiments.
2. As shown in Fig.2b in the manuscript, a sudden variation of the slope of $V_{xy}^2\omega$ is observed at $V_{BG} = 0$ V. Besides, a sudden variation of ρ_{xx} at $V_{BG} = 0$ V is also observed in Supplementary Fig.3b. What's the reason behind these phenomena? How about the reproducibility?
3. According to the analysis in Supplementary Note 1, a nonzero second order voltage could also be generated at the direction perpendicular to the ac plane. The impact of this on the measured results should be clarified.
4. To reveal the skew scattering origins of the observed nonlinear longitudinal and transverse voltages, the scaling law analysis is shown in Supplementary Note 3. However, referring to Phys. Rev. Lett. 129, 186801 (2022), the special ratio 1:-2:1 of the fitting parameters and the detailed analysis of the distributions of the skew scattering from different sources could be considered in this note.
5. The details in analysis of skew scattering mechanisms need to be reconsidered. As depicted in Line 228-229, the non-Gaussian type skew-scattering contribution is large. Considering the scaling law in Supplementary Note 3, only the fitting parameter $A2$ contains the non-Gaussian type contribution b' . And the fitting parameter $A0$ is equal to a' . However, the results of scaling law analysis in Supplementary Fig.12 shows that $A0\sigma$ and $A2\sigma^3$ have the similar values, which is in conflict with Line 228-229. Moreover, the non-Gaussian type skew-scattering contribution is neglected in Phys. Rev. Lett. 129, 186801 (2022).

Reviewer #3 (Remarks to the Author):

The authors' comprehensive point-by-point response sheds light on the intricacies of nonlinear Hall effect (NLHE) in Tellurium (Te). While the insights provided are commendable, I have a few minor comments that I'd like authors of this manuscript to address before I recommend this manuscript for publication.

Comment 5, page 17:

(Author can choose not to address this point)

Here, author has taken the example of WTe₂ to explain the symmetry restrictions and the observed NLH voltage. In WTe₂, the NLH voltage is mainly governed by the BCD, which does not allow any longitudinal 2 ω signal (i.e. no 2 ω signal when the current is applied along BCD). BCD does not contribute to longitudinal signal. However, the longitudinal can come from either artifacts or extrinsic scatterings.

Can the author comment on the following? Let's say there is no artifacts. Let's assume to significantly increase the extrinsic scatterings in WTe₂ keeping all other things like symmetries and BCD intact. Now, will there be any longitudinal 2 ω signal in WTe₂?

Comment 9, page 20:

As suggested by the author, the nonlinear Hall (NLH) voltage may align with resistivity, a correlation further supported by the resistivity data tuned through gating followed by the NLH data tuned by gating. But then, there arises a question regarding why the **total χ_{yx} data in Supplementary Fig.12 (b)** deviates from the resistivity data depicted in **Supplementary Fig. 3 (a)**?

Another related question is that, the nonlinear Hall conductivity is observed to increase with increasing carrier density, mobility and in general the linear conductivity [as shown in *NATURE NANOTECHNOLOGY* | VOL 17 | APRIL 2022 | 378–383]. The same argument is supported in *Nature Nanotechnology* 16 (4), 421-425 (2021). However, in this work, the NLH conductivity is observed to increase (NLH increases with gate) with increasing resistivity (resistivity increases with gate as shown in Supplementary Fig. 3) i.e. decreasing conductivity, which is opposite to earlier reports. Can author provide some clarification?

Author says that the NLH voltage increase might come from firstly increase of resistivity and secondly shifting of Fermi level towards band edge where the scatterings and Berry curvature might have divergence. Now, as cleared by the author, let's assume that the Te is hole doped, we contemplate the possibility of transforming it into electron-doped by elevating the temperature and applying a substantial gate voltage. This transformation could lead to contributions from the top band edge, potentially manifesting as a sign change in the nonlinear Hall voltage. Such an outcome would serve to validate the impact of Berry curvature in conjunction with scatterings. Similar measurements have been performed in *Nature* 565, 337 (2019) and *NATURE NANOTECHNOLOGY* | VOL 17 | APRIL 2022 | 378–383

Another question is regarding the gating effect on 20 nm and above thickness. I think back gating will mostly affect the bottom layers of 20 nm, however, the NLH effect is governed by only surface (bottom can also contribute but may not be that much effective). Can author clarify this point? Also, gating

effect becomes important when we reach to few layers like in *Nature* 565, 337 (2019) bilayer WTe₂ or in twisted graphene.

Responses to reviewers' reports
(Manuscript NCOMMS-23-51855A by Bin Cheng, et al.)

Detailed responses to reviewer #1:

The authors have addressed my comments in full, I recommend their publication in its current form.

Reply: We are pleased that the reviewer recognized our response. We highly appreciate the reviewer's recommendation of our manuscript.

Detailed responses to reviewer #2:

General comments: *I would like to thank the authors for their additional experiments and discussions for the giant nonlinear longitudinal and transverse nonlinearity in tellurium. Most of my questions are clearly answered, and especially I appreciate the additional careful symmetry analysis for the discussions of the surface effect.*

The following points are provided for the authors' consideration, aimed at further enhancing the quality and impact of this manuscript.

Reply: We highly appreciate the reviewer for recognizing our response. Our manuscript, particularly in terms of theoretical analysis, greatly benefits from the insightful advices from the reviewer. As detailed below, we have addressed all the newly-raised comments and suggestions, and have revised the manuscript accordingly.

Comment 1: *The angular dependence of nonlinear voltages at zero gate voltage is shown in the manuscript and Supplementary materials. However, the angular-dependent measurements with nonzero gate voltages are necessary to exclude the gate-induced nonuniformity and complete the experiments.*

Reply: Following the reviewer's suggestion, we have conducted new measurements on the angular-dependent NLHE under gate-voltage tuning, and the typical results are presented in Fig. R1. The measured second-harmonic Hall voltage $V_{xy}^{2\omega}$ shows the same one-fold angular dependence for varied V_{BG} between -20 V and 20 V, and no obvious phase shift is seen. In addition, each $V_{xy}^{2\omega}$ vs θ curve can be well fitted by the same Equation (S14) deduced from the symmetry analysis (see Supplementary Note 1).

These results indicate that the gate-induced nonuniformity, even if it exists, should have a negligible effect on the measured nonlinear Hall signal.

Figure R1 | Angular-dependent $V_{xy}^{2\omega}$ under different back-gate voltages (V_{BG}). θ is the angle between the current direction and the c-axis of Te. The solid lines are corresponding fitting curves using Equation (S14) in the Supplementary Information.

Comment 2: *As shown in Fig.2b in the manuscript, a sudden variation of the slope of $V_{xy}^{2\omega}$ is observed at $V_{BG} = 0$ V. Besides, a sudden variation of ρ_{xx} at $V_{BG} = 0$ V is also observed in Supplementary Fig.3b. What's the reason behind these phenomena? How about the reproducibility?*

Reply: We appreciate the reviewer's meticulous review. The sudden variations of both $V_{xy}^{2\omega}$ and ρ_{xx} data at $V_{BG} = 0$ V are related to the measuring process, rather than being due to some intrinsic features of the device. Taking the $V_{xy}^{2\omega}$ vs V_{BG} data for example, $V_{xy}^{2\omega}$ exhibits an overall tendency toward exponential increase as V_{BG} varies from -60 V to 45 V (see the logarithmic plot shown in the inset of Fig. 2b), and only the data at $V_{BG} = 0$ V slightly deviates from it. Given that the corresponding $V_{xy}^{2\omega}$ vs I^ω data at $V_{BG} = 0$ V and the ones under non-zero V_{BG} were obtained during two separate measurements, a little variation in the electrode contact or the measurement environment may be the cause of this deviation. This is further verified by the gate-voltage dependent data that were collected within the same measurements (e.g., the inset of Fig. 2c and Fig. 3c), wherein no sudden variation at $V_{BG} = 0$ V is observed.

Comment 3: According to the analysis in Supplementary Note 1, a nonzero second order voltage could also be generated at the direction perpendicular to the ac plane. The impact of this on the measured results should be clarified.

Reply: We agree with the reviewer that second-order Hall signals are allowed at the direction perpendicular to the ac plane. However, such an out-of-plane voltage drop should barely affect the measured $V_{xy}^{2\omega}$ data in our experiments. This is due to the fact that all the electrodes of our Te devices were fabricated within the ac plane, which will eliminate the potential mixing of the voltage signal collected from the perpendicular direction.

Motivated by this comment, we have added relevant discussions in Supplementary Note 1 in the revised manuscript.

Comment 4: To reveal the skew scattering origins of the observed nonlinear longitudinal and transverse voltages, the scaling law analysis is shown in Supplementary Note 3. However, referring to *Phys. Rev. Lett.* 129, 186801 (2022), the special ratio 1:-2:1 of the fitting parameters and the detailed analysis of the distributions of the skew scattering from different sources could be considered in this note.

Reply: We thank the reviewer for this insight comment.

For extrinsic-scattering-dominated NLHE, the presence of numerous disorder sources and their competitions make detailed analysis quite challenging. **Ref.** *Phys. Rev. Lett.* 129, 186801 (2022) is among the few reports that could separate the contributions from different types of scattering to some extent. As detailed below, this is beneficial from the specifics of both the twisted bilayer graphene (TBG) system under study and the scaling law analysis data, and is not applicable for our Te device.

(1) In **Ref. *Phys. Rev. Lett.* 129, 186801 (2022) concerning NLHE in TBG**, the general scaling equation is written as:

$$\frac{E_{xy}^{2\omega}}{(E_{xx}^{\omega})^2} = \frac{\chi_{yxx}}{\sigma} = A_0 + A_1\sigma + A_2\sigma^2 = A_0' + A_1' \frac{\sigma}{\sigma_0} + A_2' \left(\frac{\sigma}{\sigma_0}\right)^2 \quad (R1)$$

where the scaling parameters are written as:

$$\begin{aligned} A_0' &= A_0 = C^{\text{in}} + C_1^{\text{sj}} + C_{11}^{\text{sk},1} \\ A_1' &= A_1\sigma_0 = C_0^{\text{sj}} + C_{01}^{\text{sk},1} - 2C_{11}^{\text{sk},1} - C_1^{\text{sj}} \\ A_2' &= A_2(\sigma_0)^2 = C^{\text{sk},2}\sigma_0 + C_{00}^{\text{sk},1} + C_{11}^{\text{sk},1} - C_{01}^{\text{sk},1} \end{aligned}$$

Here, σ_0 is the zero-temperature conductivity. For TBG on *h*-BN substrate, the contribution of intrinsic BCD origin is prohibited by the C_3 point group symmetry, i.e.,

$C^{\text{in}} = 0$. From the reference, the extracted A_0' , A_1' , A_2' have the same orders of magnitude and identical moiré-filling-factor dependence. In particular, around the two NLH peaks, the ratios of $A_0' : A_1' : A_2'$ are both appropriately $1 : -2 : 1$, which precisely matches the ratio of the item $C_{11}^{\text{sk},1}$ in these three parameters. By combining the above characteristics, it is reasonable to conclude that $C_{11}^{\text{sk},1}$ (phonon skew scattering) dominates the scaling parameters. In comparison, the non-Gaussian type skew scattering $C^{\text{sk},2}$ and the side jump C_i^{sj} ($i = 0, 1$) are dismissed as major contributions, since they are only involved in one or two of the three parameters.

(2) **In our work on NLHE in Te**, A_0 , A_1 and A_2 in Equation (R1) were used as the parameters for the scaling law analysis (as presented in the Supplementary Note 3), since the zero-temperature conductivity σ_0 cannot be well estimated in such a semiconductor system. From the fitting results of A_0 , A_1 and A_2 , currently we can draw the conclusion that the scattering-related extrinsic mechanism plays a leading role in generating NLHE, while the intrinsic contribution (C^{in}) cannot be neglected. In addition, we have also attempted to estimate the corresponding values of A_0' , A_1' , A_2' in accordance with **Ref. Phys. Rev. Lett.** 129, 186801 (2022), by roughly estimating σ_0 through linear extrapolation of the ρ_{xx} vs T curve shown in Supplementary Fig. 3a ($\sigma_0 = 52.5 \text{ S}\cdot\text{cm}^{-1}$, which should be much higher than the actual value). A_2' , A_1' and A_0' were then calculated to be $3.379 \mu\text{mV}^{-1}$, $-8.012 \mu\text{mV}^{-1}$ and $4.765 \mu\text{mV}^{-1}$, respectively, with no specific ratio found.

Consequently, based on current results, we are unable to perform a more thorough analysis on the contributions of various scattering sources similar to the one displayed in **Ref. Phys. Rev. Lett.** 129, 186801 (2022). Nevertheless, we would like to thank the reviewer again for this comment, which motivates us to further check through our scaling law analysis. Relevant discussions have also been added in Supplementary Note 3 of the revised manuscript.

Comment 5: *The details in analysis of skew scattering mechanisms need to be reconsidered. As depicted in Line 228-229, the non-Gaussian type skew-scattering contribution is large. Considering the scaling law in Supplementary Note 3, only the fitting parameter A_2 contains the non-Gaussian type contribution b' . And the fitting parameter A_0 is equal to a' . However, the results of scaling law analysis in Supplementary Fig.12 shows that $A_0\sigma$ and $A_2\sigma^3$ have the similar values, which is in conflict with Line 228-229. Moreover, the non-Gaussian type skew-scattering contribution is neglected in Phys. Rev. Lett. 129, 186801 (2022).*

Reply: We respond to this comment in two aspects:

(1) The claim of large non-Gaussian type skew scattering contribution in Line 228-229

is supported by the scaling law analysis on etching-modified NLHE. As detailed in Supplementary Note 2, the associated scaling parameters are expressed as follows:

$$a' = C^{\text{in}} + C_0^{\text{sj}} + C_{00}^{\text{sk},1}$$

$$b' = C^{\text{sk},2}$$

On the other hand, Supplementary Note 3 presents the scaling law analysis on the temperature-dependent NLHE behaviors, and the corresponding scaling parameters are:

$$A_0 = C^{\text{in}} + C_1^{\text{sj}} + C_{11}^{\text{sk},1}$$

$$A_1 = (C_0^{\text{sj}} + C_{01}^{\text{sk},1} - 2C_{11}^{\text{sk},1} - C_1^{\text{sj}})\sigma_0^{-1}$$

$$A_2 = [C^{\text{sk},2}\sigma_0 + (C_{00}^{\text{sk},1} + C_{11}^{\text{sk},1} - C_{01}^{\text{sk},1})]\sigma_0^{-2}$$

Here the non-Gaussian type skew scattering contribution b' is indeed only contained in A_2 , just as the reviewer pointed out. However, a' is actually not equivalent to A_0 , but includes components from all three scaling parameters of A_0 , A_1 and A_2 . The conclusion of relatively large b' doesn't imply that $A_2\sigma^3$ must be larger than $A_0\sigma$, because there are many other coefficients to consider. It is thus not contradictory to the result shown in Supplementary Fig.12.

(2) In **Ref. Phys. Rev. Lett.** 129, 186801 (2022), the non-Gaussian type skew-scattering was excluded as the dominant one. However, as discussed in the reply to the reviewer's comment 4, this claim is based on the specifics of both the twisted bilayer graphene (TBG) system under study and the scaling law analysis data, especially the special ratio of 1 : -2 : 1 for the three scaling parameters under different moiré-filling factors, and is not applicable to our case of Te device.

Detailed responses to reviewer #3:

General comments: *The authors' comprehensive point-by-point response sheds light on the intricacies of nonlinear Hall effect (NLHE) in Tellurium (Te). While the insights provided are commendable, I have a few minor comments that I'd like authors of this manuscript to address before I recommend this manuscript for publication.*

Reply: We highly appreciate the reviewer for recognizing our response and recommending publication after addressing a few minor comments. As shown in the point-by-point responses below, we have addressed all the comments and suggestions raised by the reviewer, and revised the manuscript accordingly.

Comment 1: *(Author can choose not to address this point) Here, author has taken the*

example of WTe_2 to explain the symmetry restrictions and the observed NLH voltage. In WTe_2 , the NLH voltage is mainly governed by the BCD, which does not allow any longitudinal 2ω signal (i.e. no 2ω signal when the current is applied along BCD). BCD does not contribute to longitudinal signal. However, the longitudinal can come from either artifacts or extrinsic scatterings.

Can the author comment on the following? Let's say there is no artifacts. Let's assume to significantly increase the extrinsic scatterings in WTe_2 keeping all other things like symmetries and BCD intact. Now, will there be any longitudinal 2ω signal in WTe_2 ?

Reply: We thank the reviewer for this insight comment. As discussed in our manuscript, the development of the 2ω signal is stringently restricted by the crystalline symmetry. This is applicable for both the intrinsic and scattering-related extrinsic mechanisms. WTe_2 belongs to the Pm space group symmetry in light of the breaking of glide mirror symmetry M_b at the surface (*Nat. Mater.* 18, 324 (2019); *Phys. Rev. Lett.* 130, 016301 (2023)). From the as-deduced nonlinear susceptibility tensor, the longitudinal 2ω signal along the a-axis (the direction of BCD) is prohibited. Even if extrinsic scattering intensities are significantly raised while preserving the symmetry, the intrinsic symmetry constraint will remain intact and the longitudinal 2ω signal will still not emerge.

Comment 2: *As suggested by the author, the nonlinear Hall (NLH) voltage may align with resistivity, a correlation further supported by the resistivity data tuned through gating followed by the NLH data tuned by gating. But then, there arises a question regarding why the total χ_{yxx} data in Supplementary Fig.12 (b) deviates from the resistivity data depicted in Supplementary Fig. 3 (a)?*

Another related question is that, the nonlinear Hall conductivity is observed to increase with increasing carrier density, mobility and in general the linear conductivity [as shown in Nature Nanotechnology | VOL 17 | April 2022 | 378–383]. The same argument is supported in Nature Nanotechnology 16 (4), 421-425 (2021). However, in this work, the NLH conductivity is observed to increase (NLH increases with gate) with increasing resistivity (resistivity increases with gate as shown in Supplementary Fig. 3) i.e. decreasing conductivity, which is opposite to earlier reports. Can author provide some clarification?

Reply: The seemingly conflicting results between the gate-tuning and temperature-dependent NLH data actually have also been found in **Ref.** *Nat. Nanotechnol.* 17, 378 (2022), and the relevant data are presented in Fig. R2 for better demonstration. From Figs. R2a and R2b, the magnitude of NLH voltage $V_y^{2\omega}$ ($V_{xy}^{2\omega}$) and the sample resistivity ρ_{xx} have a positive correlation. However, when comparing Figs. R2c and R2d, NLH conductivity $\sigma_{yxx}^{(2)}$ (or χ_{yxx}) at $V_g = -69$ V demonstrates an inverse relationship with ρ_{xx} .

Here, we would like to emphasize that $V_{xy}^{2\omega}$ and χ_{yxx} possess different dependency behaviors on sample resistivity ρ_{xx} . For $V_{xy}^{2\omega}$, its magnitude is determined by both χ_{yxx} and ρ_{xx} in the form of $V_{xy}^{2\omega} \propto \chi_{yxx} (V_{xx}^\omega)^2 \frac{1}{\sigma} \propto (I^\omega)^2 (\rho_{xx})^3 \chi_{yxx}$. While for χ_{yxx} itself, it is determined by the magnitude of BCD (intrinsic) and/or the strength of scattering (extrinsic), and is not directly relevant to ρ_{xx} .

The above argument can be further validated by the experimental results reported in **Ref. Nat. Nanotechnol.** 16, 421 (2021). According to Fig. R2f, χ_{yxx} first decreases with increasing temperature up to ~ 150 K, and then the sign inverts and the magnitude increases up to 300 K. In contrast, sample resistivity ρ increases monotonically as the temperature increases (see Fig. R1e). Apparently, there is no direct connection between them. It has been revealed that the variation of χ_{yxx} was closely related to the change in chemical potential and, in turn, the distribution of BCD.

[REDACTED]

Figure R2 | **a**, Resistivity (ρ_{xx}) of graphene/BN superlattice versus gate voltage (V_g) measured at 1.7 K. **b**, NLH voltage ($V_y^{2\omega}$) versus V_g measured at 1.7 K. **c**, ρ_{xx} as a function of temperature measured at $V_g = -40$ V and -69 V. **d**, Nonlinear Hall conductivity $\sigma_{yxx}^{(2)}$ and the cube of mobility of graphene/BN superlattice at different temperatures for $V_g = -69$ V. **e**, Resistivity (ρ) of TaIrTe₄ versus temperature. **f**, Nonlinear Hall conductivity a function of temperature. Here, figures **a-d** are taken from **Ref. Nat. Nanotechnol.** 17, 378 (2022), and figures **e-f** are taken from **Ref. Nat. Nanotechnol.** 16, 421 (2021).

Comment 3: *Author says that the NLH voltage increase might come from firstly increase of resistivity and secondly shifting of Fermi level towards band edge where the scatterings and Berry curvature might have divergence. Now, as cleared by the author, lets assume that the Te is hole doped, we contemplate the possibility of transforming it into electron-doped by elevating the temperature and applying a substantial gate voltage. This transformation could lead to contributions from the top band edge,*

potentially manifesting as a sign change in the nonlinear Hall voltage. Such an outcome would serve to validate the impact of Berry curvature in conjunction with scatterings. Similar measurements have been performed in Nature 565, 337 (2019) and Nature Nanotechnology | VOL 17 | April 2022 | 378–383.

Reply: We agree with the reviewer that a sign change should appear in the nonlinear Hall signal when the Fermi level is shifted across the bandgap (*Phys. Rev. Lett.* 115, 216806 (2015); *Nat. Commun.* 10, 3047 (2019)), which has been experimentally demonstrated in previous studies via gate tuning. Unfortunately, we are unable to verify such a sign reversal behavior in our experiments. The as-grown Te flake samples are highly hole-doped, making it challenging to reach the electron-doped region by applying back-gate voltage across silicon oxide. In addition, applying a large positive gate voltage will also impair the electrode contact, and accordingly a well-defined quadratic dependence of $V_{xy}^{2\omega}$ on I^0 cannot be achieved.

Comment 4: *Another question is regarding the gating effect on 20 nm and above thickness. I think back gating will mostly affect the bottom layers of 20 nm, however, the NLH effect is governed by only surface (bottom can also contribute but may not be that much effective). Can author clarify this point? Also, gating effect becomes important when we reach to few layers like in Nature 565, 337 (2019) bilayer WTe₂ or in twisted graphene.*

Reply: We respond to this comment in two aspects:

- (1) In our last-round response, we have further clarified that the obtained NLHE is surface-dominated by including new findings that were inspired by the reviewer's critical comments. Here we would like to emphasize that symmetry breaking occurs at both the top and bottom surfaces, and thus both of them effectively contribute to the observed nonlinear Hall signal.
- (2) We agree with the reviewer that the effective depth for electrostatic gating is usually limited due to the screening effect. Nevertheless, for semiconductors with relatively low carrier density, gate-tuning effect is normally not confined to the bottom few layers, instead, it can extend to tens of nanometers in depth (*Semicond. Sci. Technol.* 27, 125009 (2012)). In our work, the as-grown Te flakes typically have a thickness of 20-100 nm. To ensure the gating effect, samples with a relatively small thickness of 20-30 nm were selected, and high tunability was finally achieved in both the linear- and nonlinear transport regimes.

Motivated by this comment, we have added relevant discussions in the revised manuscript (see the Methods section).

REVIEWERS' COMMENTS

Reviewer #2 (Remarks to the Author):

Considering my satisfaction with the author's response to my inquiry, I recommend publishing this paper.

Reviewer #3 (Remarks to the Author):

The author has addressed my questions/queries regarding resistivity relation to NLH effect, the gate dependence, etc. All the questions are answered satisfactorily. I recommend the revised manuscript for publication in Nature Communications.